# Recovering from Out-of-sample States via Inverse Dynamics in Offline Reinforcement Learning

**Ke Jiang**[1,2], **Jia-yu Yao**[3], **Xiaoyang Tan**[1,2*]

[1] College of Computer Science and Technology, Nanjing University of Aeronautics and Astronautics
[2] MIIT Key Laboratory of Pattern Analysis and Machine Intelligence
[3] School of Electronic and Computer Engineering, Peking University
`ke_jiang@nuaa.edu.cn`, `jiayu_yao@pku.edu.cn`, `x.tan@nuaa.edu.cn`

## Abstract

We deal with the *state distributional shift* problem commonly encountered in offline reinforcement learning during test, where the agent tends to take unreliable actions at out-of-sample (unseen) states. Our idea is to encourage the agent to follow the so called *state recovery* principle when taking actions, i.e., besides long-term return, the immediate consequences of the current action should also be taken into account and those capable of recovering the state distribution of the behavior policy are preferred. For this purpose, an inverse dynamics model is learned and employed to guide the state recovery behavior of the new policy. Theoretically, we show that the proposed method helps aligning the transited state distribution of the new policy with the offline dataset at out-of-sample states, without the need of explicitly predicting the transited state distribution, which is usually difficult in high-dimensional and complicated environments. The effectiveness and feasibility of the proposed method is demonstrated with the state-of-the-art performance on the general offline RL benchmarks.

## 1 Introduction

Reinforcement learning has made significant advances in recent years, but it has to collect experience actively to gain understanding of the underlying environments. However, such online interaction is not always practical, due to either the potential high cost of data collection procedure or its possible dangerous consequences in applications as autonomous driving or healthcare. To address these issues, offline reinforcement learning aims to learn a policy from offline datasets without doing any actual interaction with the environments [19, 13, 1].

However, directly deploying online RL algorithms, such as Deep Deterministic Policy Gradient [18], to learn the new policy from the offline dataset without proper constraints would highly likely suffer from action distributional shift due to the change in the actions generated by the new policy. This would result in the so called extrapolation error [8], i.e., the TD target could be wrongly estimated when querying those out-of-distribution (OOD) actions generated by the new policy. To address these issues, methods like Conservative Q-Learning (CQL) [16], TD3+BC [7] and Implicit Q-Learning (IQL) [14], treat the OOD actions as conterfactual queries and try to avoid performing such queries completely during learning, hence suppressing the Q-function extrapolation error. However, such pessimism for out-of-sample data could be too restricted and sample inefficient, as not all out-of-sample(unseen) states are not generalizable [20].

To effectively generalize to out-of-sample or even OOD states, it is necessary to constrain the behavior of agent at those states, otherwise the policy extrapolation error that occurs at test stage may drive the

---

[*]Corresponding Author

37th Conference on Neural Information Processing Systems (NeurIPS 2023).

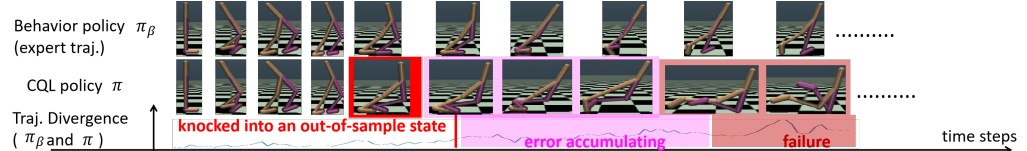

Figure 1: An example of CQL agent's failure for the accumulative error due to state distributional shift on a Walker2d robotic agent. The interval between every two images is ten steps.

agent's transited state distribution away from the offline dataset, referred as the *state distributional shift* problem - a problem has not been paid enough attention as most previous works focus on addressing the issues of OOD actions during training [16, 7]. The problem of *state distributional shift* is likely to arise in those extensive and non-stationary environments, where it is impossible for the agent to cover the entire state space during training. A brief example is illustrated in Figure 1, where a robotic agent, trained using offline RL frameworks like CQL, is knocked by some external force unintentionally and falls into an out-of-sample state during test, which leads to failure finally due to the unreliable actions taken at this out-of-sample(unseen) state [3] and the error confounding effects.

To deal with the above problem, a natural idea is to teach the agent to recover from these out-of-sample states to their familiar regions, i.e., the demonstration of offline datasets. For this purpose, we propose the Out-of-sample State Recovery (OSR) method to implement this *state recovery* principle. Our idea is to estimate the actions whose consequence would be in the support region of the offline dataset using inverse dynamics model (IDM) [2]. The estimated IDM can be interpreted as an extended policy with pre-defined consequences, hence being suitable for guiding the learning of the new policy. Theoretically, we show that the proposed OSR makes the transited state distribution of the learnt policy at out-of-sample states align well with the offline dataset, without the need of building any forward dynamics models. We also propose a modified OSR-v that suppresses the probability of selecting actions according to the risk of OOD consequences through imposing extra value-constraints when decision making. We experimentally[2] demonstrated that the proposed OSR outperforms several closely related state-of-the-art methods in offline AntMaze/MuJoCo control suites with various settings.

In what follows, after an introduction and a review of related works, Section 3 provides a concise introduction to the background of offline RL and our work. The OSR method is presented in detail with theoretical analysis of its effectiveness in Section 4.2. The variant of OSR, called OSR-v, is then introduced with theoretical analysis of its advantages in Section 4.3. In Section 4.4, the practical implementation of both methods is described, including the loss function used. Experimental results are presented in Section 5 to evaluate the effectiveness of both methods under various settings. Finally, the paper concludes with a summary of the findings and contributions.

## 2    Related work

**Robust offline reinfocement learning.**    Robustness is critical for offline RL methods to implement in real-world applications. Recent work RORL [25] adds smoothing term to relax the conservatism of algorithms like Conservative Q-Learning, making the agent generalize to out-of-sample states. ROMI [23] introduce reverse mechanism to build links between isolated data clusters based on out-of-sample data. However, it is still possible that the agent would drift away from the dataset as the results of the well-known error compounding effects due to the myopic of the new policy, referred as *state distributional shift* problem, which is more important in practical applications. For example, in healthcare [12], it is critical for the agent to make decisions with safe consequences, i.e., generating reliable trajectories. Theoretically, it is shown that control the divergence of the learnt policy from the behavior policy helps to bound the *state distributional shift* [22], but the bound is actually very loose [15]. Recently model-based State Deviation Correction (SDC) [27] builds a dynamics model and a transition model to guide the agent to generate in-distribution trajectories, through which to enhance the agent's safety and controllability. However, building a high-capacity dynamics model is not always practical in complicated applications, highlighting the necessity to constrain the agent's behavior without constructing any forward model to predict the high-dimensional observations.

---

[2]Our code is available at https://github.com/Jack10843/OSR

**Inverse dynamics model.** An inverse dynamics model $I(a|s', s)$ predicts the action distribution that explains the transition between a given pair of states. Inverse dynamics models haven been applied to improving generalization to real-world problems [6], Markov representation learning [2], defining intrinsic rewards for exploration [5]. In our work, the inverse dynamics model is interpreted as a consequence-constraint policy that generates action distributions with predictable consequences, which in turn plays the role of supervision to the new policy.

## 3 Background

A reinforcement learning problem is usually modeled as a Markov Decision Process (MDP), which can be represented by a tuple of the form $(S, A, P, R, \gamma)$, where $S$ is the state space, $A$ is the action space, $P$ is the transition probability matrix, $R$ and $\gamma$ are the reward function and the discount factor. A policy is defined as $\pi : S \rightarrow A$ and trained to maximize the expected cumulative discounted reward in the MDP:

$$\max_\pi \mathbb{E}\big[\sum_{t=0}^\infty \gamma R(s_t, \pi(a_t|s_t))\big] \tag{1}$$

In general, we define a Q-value function $Q^\pi(s, a) = \mathbb{E}[\sum_{t=0}^\infty \gamma R(s_t, \pi(a_t|s_t))|s, a]$ to represent the expected cumulative rewards. Q-learning is a classic method that trains the Q-value function by minimizing the Bellman error over $Q$ [24]. In the setting of continuous action space, Q-learning methods use exact or an approximate maximization scheme, such as CEM [11] to recover the greedy policy, as follows,

$$Q \leftarrow \arg\min_Q \mathbb{E}\big[R(s, a) + \gamma \mathbb{E}_{a' \sim \pi(\cdot|s')} Q(s', a') - Q(s, a)\big]^2$$
$$\pi \leftarrow \arg\max_\pi \mathbb{E}_s \mathbb{E}_{a \sim \pi(\cdot|s)} Q(s, a) \tag{2}$$

In offline setting, Q-Learning algorithms learns a Q-value function $Q^\pi(s, a)$ and a policy $\pi$ from a dataset $\mathcal{D}$, which is collected by a behavior policy $\pi_\beta$. Since there is always an action distributional shift of the new policy $\pi$ and the behavior policy $\pi_\beta$, this basic recipe fails to estimate the Q-values for OOD state-action pairs. Conservative Q-Learning (CQL), as a representative OOD-constraint offline RL algorithm, tries to underestimate the Q-values for OOD state-action pairs to prevent the agent from extrapolation error [16]. The CQL term is as follows,

$$\min_Q \big[\mathbb{E}_{s \sim \mathcal{D}, a \sim \pi(\cdot|s)} Q(s, a) - \mathbb{E}_{s, a \sim \mathcal{D}} Q(s, a)\big] \tag{3}$$

However, at test stage, there still exists a static *state distributional shift* of the new policy $\pi$ and the behavior policy $\pi_\beta$, which may accumulate the discrepancy between the agent and demonstration of the dataset. To migrate this, State Deviation Correction (SDC) [27] aims to train a policy choosing actions whose visited states are as closer to the dataset as possible. Specifically, the SDC trains a dynamics model $M$ and a transition model $U$, to optimize the new policy, as follows,

$$\min_\pi \lambda \cdot \mathbb{E}_{s \sim \mathcal{D}} D\big(M(\cdot|\hat{s}, \pi(\cdot|\hat{s})), U(\cdot|s)\big) \tag{4}$$

where $\hat{s}$ is a perturbed version of the original state $s$, $\lambda > 0$ is the weight for the SDC regularization, and $D$ is a distance measure, which is maximum mean discrepancy (MMD) in [27].

## 4 Out-of-sample state recovery using inverse dynamics

In this section, we first describe our Out-of-sample State Recovery (OSR) method in detail, as a policy regularization onto the actor loss, which is introduced in detail in section 4.2. Then we also propose a variant of OSR, OSR via value-constraints (OSR-v), for the purposes to reduce the hyperparameters and test the feasibility of practising *state recovery* principle in different modes, in section 4.3. Finally, we specifically describe how to implement OSR and OSR-v in practice in section 4.4.

### 4.1 Noise injection onto dataset

It is well-known that noise injection is helpful to address the covariate shift problem in deep learning [17]. While this technique is adopted in [27] to simulate the OOD states, in this work we interpret

the states perturbed with linear Gaussian noise as counter-examples on how to recover from those states into in-sample states. Given an offline dataset $\mathcal{D}$, which consists of quadruples $(s, a, r, s')$, we first perform data augmentation onto it to form a mixed dataset.

Specifically, given a quadruple $(s, a, r, s')$, we perturb the state $s$ to obtain a noisy state,

$$\hat{s} = s + \beta \cdot \epsilon \tag{5}$$

where $\epsilon$ is sampled from the standard Gaussian distribution $\mathcal{N}(0, 1)$ and $\beta$ is usually set to be a small constant. Besides, we could also utilize more complex methods to perturb the states, such as the adversarial attacks in [25]. In this way, we obtain a set of perturbed quadruples $(\hat{s}, a, r, s')$, and group them into a new perturbed dataset $\tilde{\mathcal{D}}$. Note that we do not perturb the next state $s'$, to preserve the destination we wish the agent to jump from $\hat{s}$.

Finally we combine the two datasets $\mathcal{D}$ and $\tilde{\mathcal{D}}$ to form a new dataset $\mathcal{D}_{tot}$, i.e., $\mathcal{D}_{tot} = \mathcal{D} + \tilde{\mathcal{D}}$, where each element in $D_{tot}$ is denoted as $(\tilde{s}, a, r, s')$. Next we describe how to train a inverse dynamics model to learn how to safely perform transition from $\{\tilde{s}\}$ to $s'$.

### 4.2   Learning to recover from out-of-sample states via policy-constraints

A inverse dynamics model (IDM), denoted as $I^{\pi_\beta}(a|s, s')$, is defined in terms of the behavior policy $\pi_\beta$, dynamics model $P(s'|s, a)$ and the transition function $P(s'|s, \pi_\beta)$ via Bayes' theorem, as follows,

$$I^{\pi_\beta}(a|s, s') = \frac{\pi_\beta(a|s)P(s'|s, a)}{P(s'|s, \pi_\beta)} \tag{6}$$

where $P(s'|s, \pi_\beta) = \sum_{a \in A} P(s'|s, a)\pi_\beta(a|s)$. Note that to learn a IDM model, we don't have to estimate the dynamic model of the environment but only needs to treat this task as a usual function approximation problem using a neural network. In particular, using the quadruples $(\tilde{s}, a, r, s')$ in $\mathcal{D}_{tot}$, the needed IDM $I^{\pi_\beta}(a|s, s')$ can be estimated using a probabilistic regression model with $\tilde{s}, s'$ as input and action $a$ as output.

With the estimated IDM model available, we interpreted it as an extended policy with desired immediate consequence known, illustrating how to recover from out-of-samples into in-sample(safe) states, then use it to guide the learning of the new policy. In this work, we use the Kullback-Leibler(KL) divergence to measure the divergence between two distributions. In particular, we minimize the KL divergence between $I^{\pi_\beta}(a|\tilde{s}, s')$ and the new policy $\pi(a|\tilde{s})$, as follows,

$$\min_{\pi} \mathbb{E}_{\tilde{s} \sim \mathcal{D}_{tot}} KL\Big(\mathbb{E}_{s' \sim P(s'|s, \pi_\beta)} I^{\pi_\beta}(a|\tilde{s}, s') \Big\| \pi(a|\tilde{s})\Big) \tag{7}$$

where the $\tilde{s}$ is sampled from the mixed dataset $\mathcal{D}_{tot}$ and $s'$ is its in-sample consecutive state. (7) is the OSR term aiming to play the same role of traditional model-based SDC, but without modeling the high-dimensional observations. Obviously, (7) utilizes a forward KL divergence, in which the difference between $I^{\pi_\beta}(a|\tilde{s}, s')$ and $\pi(a|\tilde{s})$ is weighted by $I^{\pi_\beta}(a|\tilde{s}, s')$. Minimizing this forward KL divergence is equivalent to maximizing the policy likelihood as, $\max_{\pi} \mathbb{E}_{\tilde{s} \sim \mathcal{D}_{tot}, s' \sim P(s'|s, \pi_\beta), a \sim I^{\pi_\beta}(a|\tilde{s}, s')}[\log \pi(a|\tilde{s})]$, which is easy to achieve by the definition of KL divergence. In this manner, the learnt policy $\pi$ is able to recover the average behavior of the IDM $E_{s' \sim P(s'|s, \pi_\beta)} I^{\pi_\beta}(a|\tilde{s}, s')$ with greatest probability.

To further explain the feasibility of OSR term, Theorem 1 shows that the OSR term aligns the transited state distribution of the learnt policy with the offline dataset at out-of-sample states in the setting of bounded action space, which is enough to meet the requirements of most RL environments. First, two assumptions are given to serve Theorem 1 in Appendix A, where the assumptions give the continuity and positivity for the transition function of $\pi_\beta$, and the positivity for the new policy we train. Then,

**Theorem 1.** *Given two consecutive state $s$ and $s'$, the out-of-sample state $\hat{s}$ within the $\epsilon$-neighbourhood of $s$ and the behavior policy $\pi_\beta$. Given an inverse dynamics model $I^{\pi_\beta}(a|s, s')$. In bounded action space setting, the following two optimization formulas are equivalent,*

$$\min_{\pi} KL\Big(\mathbb{E}_{s' \sim P(s'|s, \pi_\beta)} I^{\pi_\beta}(a|\hat{s}, s') \Big\| \pi(a|\hat{s})\Big) \iff \min_{\pi} KL\Big(P(s'|s, \pi_\beta) \Big\| P(s'|\hat{s}, \pi)\Big)$$

*, that is, the two optimization formulas achieve the same policy.*

The proof of Theorem 1 is conducted in Appendix B.1. It theoretically guarantees that OSR have the effect that the learnt policy is regularized to recover the transited state distribution $P(s'|\hat{s}, \pi)$ at out-of-sample state $\hat{s}$ aligned well with the transited state distribution $P(s'|s, \pi_\beta)$ of the behavior policy $\pi_\beta$ at the corresponding in-sample state $s$. For the fact that the offline dataset is generated via the behavior policy $\pi_\beta$, then $P(s'|s, \pi_\beta)$ is the in-distributional transited state distribution of the offline dataset. Therefore, we remark that OSR helps aligning the new policy's transited state distribution $P(s'|\hat{s}, \pi)$ at out-of-sample state $\hat{s}$ with the offline dataset. Interestingly, we find that the term $\min_\pi KL\big(P(s'|s, \pi_\beta)\big\|P(s'|\hat{s}, \pi)\big)$ could also be termed as a KL-divergence version of SDC introduced in (4).

### 4.3 Value-constraint out-of-sample state recovery: a variant

In this section, we propose Out-of-sample State Recovery via value-constraints (OSR-v) as a variant of OSR proposed before. The proposed OSR-v penalizes the Q-values of those actions that prefer transiting to OOD states at out-of-sample states, which drives the agent able to choose the actions that transits to in-sample states, recovering from out-of-sample states. In particular, given two consecutive states $\tilde{s}$ and $s'$ sampled from the mixed dataset $\mathcal{D}_{tot}$ constructed before, we overestimate the Q-values of actions that transits to in-sample states, i.e., actions generated by the IDM $I^{\pi_\beta}(\cdot|\tilde{s}, s')$, while suppressing the Q-values of actions with unknown consequences, i.e., actions generated by the new policy $\pi$, as follows,

$$\min_Q \mathbb{E}_{\tilde{s}, s' \sim \mathcal{D}_{tot}} \Big( \mathbb{E}_{\hat{a} \sim \pi(\cdot|\tilde{s})} Q(\tilde{s}, \hat{a}) - \mathbb{E}_{\hat{a}_{tar} \sim I^{\pi_\beta}(\cdot|\tilde{s}, s')} Q(\tilde{s}, \hat{a}_{tar}) \Big) \qquad (8)$$

where $\hat{a}_{tar}$ refers to the action sampled from IDM, which is referred as target action.

In practice, we observe that OSR-v enable the agent to follow the *state recovery* principle as well. In order to rigorously explain this phenomenon, we give Theorem 2 under the assumptions mentioned in Appendix A to show that performing OSR-v is equivalent to underestimating the value of OOD states while overestimating the value of in-sample states, which guides the agent prefer to transit to those in-sample(safe) states, avoiding error accumulated via *state distributional shift*.

**Theorem 2.** *Given two consecutive state $s$ and $s'$, the out-of-sample state $\hat{s}$ within the $\epsilon$-neighbourhood of $s$ and the behavior policy $\pi_\beta$. $\pi$ is the new policy. Define $Q(s, a)$ as a approximal Q-value function and $V(s)$ as a approximal value function. Given an inverse dynamics model $I^{\pi_\beta}(a|s, s')$. Then the following two optimization formulas are approximately equivalent:*

$$\min_Q \mathbb{E}_{\hat{a} \sim \pi(\cdot|\hat{s})} Q(\hat{s}, \hat{a}) - \mathbb{E}_{\hat{a}_{tar} \sim I^{\pi_\beta}(\cdot|\hat{s}, s')} Q(\hat{s}, \hat{a}_{tar}) \qquad (9)$$

$$\overset{\sim}{\Leftrightarrow} \min_V \mathbb{E}_{s' \sim P(s'|\hat{s}, \pi)} V(s') - \mathbb{E}_{s' \sim P(s'|s, \pi_\beta)} V(s') \qquad (10)$$

*, that is, they achieve the same policy approximately. Then the difference of the two optimization formulas is no larger than $2\delta \sum_a \pi_\beta(a|\hat{s}) Q(\hat{s}, a)$.*

The proof of Theorem 2 is conducted in Appendix B.4. Theorem 2 demonstrates the approximate equivalence of (9) and (10). There is actually a little gap between (9) and (10) which is caused by the existence of the continuous gap $\delta$, so when $\delta$ is large, (9) and (10) are quite different. However, if we assume the infinite norm of reward function $\|R\|_\infty \leq 1$, then $2\delta \sum_a \pi_\beta(a|\hat{s}) Q(\hat{s}, a) \leq \frac{2\delta}{1-\gamma}$, where $\gamma$ is the discount factor, and by the conservative selection of the magnitude of the noise $\epsilon$, the $\delta$ would be a quite tiny value in practice.

The $s'$ sampled from $P(s'|s, \pi_\beta)$, the state distribution aligned with the offline dataset $\mathcal{D}$, is in-sample, then overestimating of such $s'$ increases the agent's preference to transiting to in-sample states, which is supported by the reasoning presented in Proposition 1, as follows,

**Proposition 1.** *Given an arbitrary state $s$ and its target state $s'$. $Q$ is an approximal Q-value function and $V$ is its value function. $\pi_Q$ is the learnt policy according to Q. Let $\pi_Q(a|s)$ is positive correlated to $Q(s, a)$, noted as $\pi_Q(a|s) \propto Q(s, a)$, and assume $\exists a, P(s'|s, a) > 0$, then $P(s'|s, \pi_Q) \propto V(s')$.*

The proof and more discussion of Proposition 1 is conducted in Appendix B.2. For the fact that out-of-sample(noisy) state $\hat{s}$ is close to $s$, the in-sample prior of $s'$, so we assume $\exists a, P(s'|\hat{s}, a) > 0$, and via Proposition 1, $\mathbb{E}_{s' \sim P(s'|s, \pi_\beta)} P(s'|\hat{s}, \pi) \propto E_{s' \sim P(s'|s, \pi_\beta)} V(s')$. Therefore, the result of Theorem 2 and Proposition 1 guarantees that OSR-v can align the transited state distribution of the new policy with the offline dataset, avoiding compounding *state distributional shift* as well.

## 4.4 Implementation and algorithm summary

We implement our methods, OSR and OSR-v, introduced in previous sections based on an Conservative Q-Learning(CQL) framework to practice the *state recovery* principle in the setting of countinous action space, without modeling the dynamics of the environment. To be specific, we would implement OSR in policy-constraint mode and OSR-v in value-constraint mode.

**Policy-constraint mode (OSR).** The policy-constraint based method, OSR proposed in (7), could be implemented as SR loss $L_{sr}$ in (11), where the detailed derivation is presented in Appendix B.5.

$$L_{sr} = E_{\tilde{s},s' \sim \mathcal{D}_{tot}} KL\Big(I^{\pi_\beta}(a|\tilde{s},s')\Big\|\pi(a|\tilde{s})\Big) \tag{11}$$

Then the problem is to estimate the KL divergence in (11) under the condition of continuous action space. In general, we often model the policy as $\pi(a|\tilde{s}) = \mathcal{N}(\mu_\pi, \sigma_\pi; \tilde{s})$ and the IDM as $I^{\pi_\beta}(a|\tilde{s},s') = \mathcal{N}(\mu_I, \sigma_I; \tilde{s}, s')$, where $\mathcal{N}$ notes Gaussian distribution. First, we construct the SR loss for the policy network to estimate the KL divergence mentioned in (11).

There is an analytical solution for calculating the KL divergence between two Gaussian distributions, the Gaussian policy $\pi(a|\tilde{s}) = \mathcal{N}(\mu_\pi, \sigma_\pi; \tilde{s})$ and the Gaussian IDM $I^{\pi_\beta}(a|\tilde{s},s') = \mathcal{N}(\mu_I, \sigma_I; \tilde{s}, s')$, so we can transfer (11) as the following SR loss:

$$L_{sr} = \mathbb{E}_{\tilde{s},s' \sim \mathcal{D}_{tot}} \Big[\sum_{d=1}^{D}(\log \frac{\sigma_I^d}{\sigma_\pi^d} + \frac{(\sigma_I^d)^2 + (\mu_I^d - \mu_\pi^d)^2}{2(\sigma_\pi^d)^2})\Big] \tag{12}$$

where subscript $d$ represents the value of $d^{th}$ dimension of the $D - dimensional$ variable. After constructing the term of OSR in the settings of both discrete and continuous action space, we construct the actor loss function of the policy network as follows,

$$L_\pi = \mathbb{E}_{\tilde{s} \sim \mathcal{D}_{tot}, \hat{a} \sim \pi(\cdot|\tilde{s})}\Big[Q(\tilde{s}, \hat{a})\Big] + \lambda L_{sr}$$

where $\lambda$ is the balance-coefficient and the $L_{sr}$ is as (12) in the setting of continuous action space. $L_{sr}$ imply a behavior cloning from the target distribution estimated via the inverse dynamics model $I^{\pi_\beta}(\cdot|\tilde{s}, s')$ to our new policy to enable the agent always transits to those in-sample states $s'$ from no matter in-sample $s$ or out-of-sample $\hat{s}$, i.e., $\tilde{s}$ from the mixed dataset $\mathcal{D}_{tot}$ mentioned in section 4.1. Then the critic loss function for the Q-networks is,

$$L_Q = E_{s,a,r,s' \sim \mathcal{D}}\Big[\alpha \cdot \big(\mathbb{E}_{\hat{a} \sim \pi(\hat{a}|s)}Q(s, \hat{a}) - Q(s, a)\big) + \big(r + \gamma \mathbb{E}_{a' \sim \pi(\cdot|s')}Q(s', a') - Q(s, a)\big)^2\Big]$$

where $\alpha$ is the balance-coefficient; The second term is an one-step Bellman error, as (2) shows.

**Value-constraint mode (OSR-v).** We note the term of (8) mentioned in section 4.3, as $L_{sr-v}$, to regularize the Q-networks. Then the critic loss function for the Q-networks is as follows,

$$L_Q = \alpha \cdot L_{sr-v} + \mathbb{E}_{s,a,r,s' \sim \mathcal{D}}\Big[r + \gamma \mathbb{E}_{a' \sim \pi(\cdot|s')}Q(s', a') - Q(s, a)\Big]$$

$$where \quad L_{sr-v} = \mathbb{E}_{\tilde{s},s' \sim \mathcal{D}_{tot}}\Big(\mathbb{E}_{\hat{a} \sim \pi(\cdot|\tilde{s})}Q(\tilde{s}, \hat{a}) - \mathbb{E}_{\hat{a}_{tar} \sim I^{\pi_\beta}(\cdot|\tilde{s},s')}Q(\tilde{s}, \hat{a}_{tar})\Big)$$

where $\alpha$ is the balance-coefficient. We replace the CQL regularization, because $L_{sr-v}$ would degrade to CQL regularization when it performs on the in-sample part(the original offline dataset $\mathcal{D}$) of the mixed dataset $\mathcal{D}_{tot}$, as Proposition 2 shows.

**Proposition 2.** *When training on the original dataset $\mathcal{D}$, $L_{sr-v}$ is equivalent to CQL regularization.*

The proof of Proposition 2 is conducted in Appendix B.3. Proposition 2 shows the feasibility of replacing the term of CQL by the OSR term, so that fewer hyperparameters are introduced. $L_{sr-v}$ overestimates the actions that prefer transiting to in-sample sates to avoid error accumulation. Then the actor loss $L_\pi$ we use in OSR-v is as follows,

$$L_\pi = \mathbb{E}_{\tilde{s} \sim \mathcal{D}_{tot}, \hat{a} \sim \pi(\cdot|\tilde{s})}\Big[Q(\tilde{s}, \hat{a})\Big]$$

To sum up, both OSR and OSR-v optimize actor loss $L_\pi$ to update the policy network $\pi$ and critic loss $L_Q$ to update our Q-networks, and output the learnt policy network $\pi$. The whole process of OSR is summarized in Algorithm 1 in Appendix C while the whole process of OSR-v is summarized as Algorithm 2 in Appendix C.

Table 1: Results of **OSR(ours), OSR-v(ours)**, SDC, RAMBO, MOPO, IQL and CQL on offline MuJoCo and AntMaze control tasks averaged over 4 seeds. * indicates the average without 'expert' datasets. The top-2 highest scores in each benchmark are bolded.

| Task name | CQL | IQL | MOPO | RAMBO | SDC | OSR | OSR-v |
|---|---|---|---|---|---|---|---|
| Halfcheetah-r. | 35.4 | 13.1 | 31.9 | **40.0** | **36.2** | 35.2± 1.2 | 35.9± 1.0 |
| Walker2d-r. | 7.0 | 5.4 | 13.3 | 11.5 | **14.3** | 13.5± 2.8 | **14.1**± 3.9 |
| Hopper-r. | 10.8 | 7.9 | 13.0 | **21.6** | 10.6 | 10.3± 3.1 | **13.5**± 3.4 |
| Halfcheetah-m. | 44.4 | 47.4 | 40.2 | **77.6** | 47.1 | 48.8± 0.2 | **49.1**± 0.4 |
| Walker2d-m. | 79.2 | 78.3 | 26.5 | **86.9** | 81.1 | **85.7**± 0.6 | 85.1± 0.7 |
| Hopper-m. | 58.0 | 66.2 | 14.0 | **92.8** | 91.3 | 83.1± 1.9 | **95.2**± 1.1 |
| Halfcheetah-m.-r. | 46.2 | 44.2 | **54.0** | **68.9** | 47.3 | 46.8± 0.4 | 48.5± 0.6 |
| Walker2d-m.-r. | 26.7 | 73.8 | **92.5** | 85.0 | 30.3 | **87.9**± 1.1 | 87.8± 1.0 |
| Hopper-m.-r. | 48.6 | 94.7 | 42.7 | 96.6 | 48.2 | **96.7**± 1.7 | **101.2**± 0.8 |
| Halfcheetah-m.-e. | 62.4 | 86.7 | 57.9 | 93.7 | **101.3** | 94.7± 0.9 | **99.1**± 0.4 |
| Walker2d-m.-e. | 98.7 | 109.6 | 51.7 | 68.3 | 105.3 | **114.3**± 1.5 | **112.9**± 0.8 |
| Hopper-m.-e. | 111.0 | 91.5 | 55.0 | 83.3 | 112.9 | **113.1**± 0.6 | **113.2**± 0.6 |
| Halfcheetah-e. | **104.8** | 95.0 | - | - | **106.6** | 97.7± 0.7 | 102.1± 0.8 |
| Walker2d-e. | **153.9** | 109.4 | - | - | 108.3 | 110.3± 0.4 | **111.4**± 0.8 |
| Hopper-e. | 109.9 | 109.9 | - | - | 112.6 | **113.1**± 0.5 | 112.9± 0.4 |
| **MuJoCo-v2 Avg.** | 66.5 | 68.9 | 41.1* | 68.5* | 70.2 | **76.7(69.2*)** | **78.9(71.5*)** |
| AntMaze-u. | 74.0 | 87.5 | 0.0 | 25.0 | 89.0 | **89.9**±1.9 | **92.0**±0.8 |
| AntMaze-m.-p. | 61.2 | 71.2 | 0.0 | 16.4 | **71.9** | 66.0±2.5 | **71.3**±1.3 |
| AntMaze-l.-p. | 15.8 | **39.6** | 0.0 | 0.0 | 37.2 | 37.9±2.7 | **38.3**±2.5 |
| AntMaze-u.-d. | **84.0** | 62.2 | 0.0 | 0.0 | 57.3 | **74.0**±2.8 | 69.0±2.5 |
| AntMaze-m.-d. | 53.7 | 70.0 | 0.0 | 23.2 | **78.7** | **80.0**±2.7 | 77.0±2.4 |
| AntMaze-l.-d. | 14.9 | **47.5** | 0.0 | 2.4 | 33.2 | **37.9**±1.5 | 34.0±2.2 |
| **AntMaze-v0 Avg.** | 50.6 | 63.0 | 0.0 | 22.3 | 61.2 | **64.3** | **63.4** |

# 5 Experiments

In experiments we aim to answer: 1) Does the proposed OSR help to achieve the state-of-the-art performance in offline RL? 2) Is the inverse dynamics model able to guide the policy recovering from out-of-sample states? 3) Is OSR able to improve the agent's robustness for out-of-sample situations? 4) Does OSR generalize well if the offline dataset only covers limited valuable area of the state space. Our experiments are organized as follows: Firstly, we conduct a comparative study on the MuJoCo and AntMaze benchmarks in the D4RL datasets for different versions of our method; then we design an out-of-sample MuJoCo setting to evaluate the generalization ability of CQL, SDC, the proposed OSR and OSR-v, and explore the behavior of the inverse dynamics model(IDM) and OSR policy; finally, we test the efficacy of OSR when presented with limited valuable data. A brief introduction of our code is available in Appendix D.9. **MuJoCo.** There are three types of high-dimensional control environments representing different robots in D4RL: Hopper, Halfcheetah and Walker2d, and five kinds of datasets: 'random', 'medium', 'medium-replay', 'medium-expert' and 'expert'. The 'random' is generated by a random policy and the 'medium' is collected by an early-stopped SAC [9] policy. The 'medium-replay' collects the data in the replay buffer of the 'medium' policy. The 'expert' is produced by a completely trained SAC. The 'medium-expert' is a mixture of 'medium' and 'expert' in half. **AntMaze.** AntMaze in D4RL is a maze-like environment where an ant agent must navigate through obstacles to reach a goal location. There are three different layouts of maze: 'umaze', 'medium' and 'large', and different dataset types: 'fixed', 'play' and 'diverse' which has different start and goal locations used to collect the dataset. **Hyperparameter Details.** We base the hyperparameters for OSR on those used in CQL [16]. We observe the performance of OSR is influenced by the noise magnitude $\beta$ and SR weighting $\lambda$. *Therefore, we conduct a sensitivity analysis of OSR in Appendix D.1.*

*Other experiments, including Sensitive Analysis, Ablation study, Comparison Study of implementations with Q-ensemble trick and so on, are attached in Appendix D.*

## 5.1 Comparative study on offline MuJoCo/AntMaze suite

In this section, we compare our methods with several significant methods, including MOPO [26], RAMBO [21], IQL [14], CQL [16] and SDC [27], on the D4RL dataset in the MuJoCo benchmarks. Part of the results for the comparative methods are obtained by [27, 21]. All these methods are implemented without Q-ensemble trick [4] and results are shown in Table 1, where the proposed OSR and OSR-v achieve the similar or better performance than other methods on most tasks and the best average score, and we also remark that OSR and OSR-v generalize significantly better on the 'medium-expert' datasets where limited high-valuable data is available. More training details of OSR/OSR-v could be achieved in Appendix D.3. Besides, we also implement OSR with Q-ensemble trick of 10 Q-functions [4], termed as OSR-10, to compare with Q-ensemble based methods like RORL [25], SAC-10 and EDAC-10 [4], in Appendix D.4. From the results above, we conclude that OSR helps to achieve the state-of-the-art performance in offline RL.

## 5.2 Out-of-sample MuJoCo setting

To evaluate whether the inverse dynamics model and the proposed OSR policy enable the agent to recover from out-of-sample situations, we design Out-of-sample MuJoCo (OOSMuJoCo) benchmarks by adding different steps of random actions (Gaussian noise with magnitude of 1e-3) to simulate external force to knock the agent into out-of-sample states on Halfcheetah, Walker2d and Hopper. There are 3 levels of external force: slight(s) takes 5 steps of random actions, moderate(m) takes 10 steps and large(l) takes 20 steps. The visualization of OOSMuJoCo is shown in Figure 2.

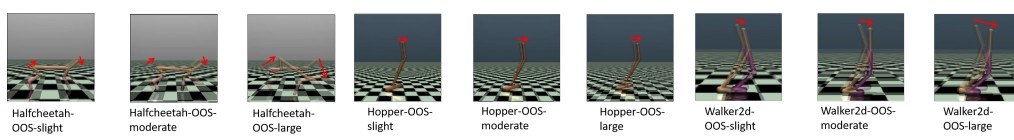

Figure 2: The visualizations of the 9 OOSMuJoCo benchmarks. The red arrows represent the direction the agents move after external force, and the agents fall into out-of-sample states then.

We evaluate the performance of policies trained by CQL, SDC, OSR and OSR-v on 'medium-expert' datasets in the three benchmarks. The score and performance decrease of these policies across the 9 OOSMuJoCo benchmarks are shown in Table 2, where the performance decrease is measured as the percentage reduction in scores from OOSMuJoCo compared to the results of standard MuJoCo environments in Table 1. Our results indicate that the proposed OSR/OSR-v experiences significantly less performance degradation than the other two methods across most of the OOSMuJoCo benchmarks, demonstrating better robustness against perturbation for out-of-sample situations in the complicated and non-stationary real-world environments.

Table 2: Results of CQL, SDC, OSR/OSR-v in OOSMuJoCo setting on the normalized return and decrease metric averaged over 4 seeds. The highest score and lowest decrease are bolded.

| Task name | CQL score | dec.(%) | SDC score | dec.(%) | OSR score | dec.(%) | OSR-v score | dec.(%) |
|---|---|---|---|---|---|---|---|---|
| Halfcheetah-OOS-s. | 56.2 | 10.0 | **100.9** | 0.4 | 94.1 | **0.6** | 98.4 | 0.7 |
| Halfcheetah-OOS-m. | 52.9 | 15.3 | **99.8** | 1.5 | 92.7 | 2.1 | 98.4 | **0.7** |
| Halfcheetah-OOS-l. | 42.1 | 32.5 | 83.3 | 17.8 | **91.7** | **3.2** | 79.8 | 19.5 |
| Walker2d-OOS-s. | 81.1 | 47.3 | 102.7 | 2.5 | **113.3** | **0.9** | 110.3 | 2.3 |
| Walker2d-OOS-m. | 77.4 | 21.6 | 102.2 | 2.9 | **113.2** | **1.0** | 110.9 | 1.8 |
| Walker2d-OOS-l. | 52.0 | 47.3 | 95.6 | 9.2 | **110.1** | **3.7** | 106.7 | 5.5 |
| Hopper-OOS-s. | 110.4 | **0.6** | 112.0 | **0.6** | 111.4 | 1.5 | **112.5** | **0.6** |
| Hopper-OOS-m. | 97.3 | 12.3 | 110.8 | 1.9 | 109.2 | 3.4 | **112.2** | **0.9** |
| Hopper-OOS-l. | 54.4 | 51.0 | 89.2 | 20.9 | **106.1** | **6.2** | 105.6 | 6.7 |

Besides, as a toy example, we visualize the Halfcheetah agent within an expert trajectory produced by the behavior policy $\pi_\beta$ on the offline dataset $\mathcal{D}$, as is shown in Figure 3, then we also visualize the trajectories generated by the policy of OSR and the inverse dynamics model(IDM) guided from

the behavior policy, where we set the target state of IDM to the corresponding state in the expert trajectory. We add external force to knock the agent into an out-of-sample state and we observe that the agents equiped with IDM and OSR policy are both able to recover from the out-of-sample states. The state recovery processes of other 2 benchmarks is shown in Appendix D.5.2 and the visualizations of state distribution is shown in Appendix D.5.1. The results above demonstrate that the the inverse dynamics model is able to guide the policy recovering from out-of-sample state and OSR is able to improve the agent's robustness for out-of-sample situations.

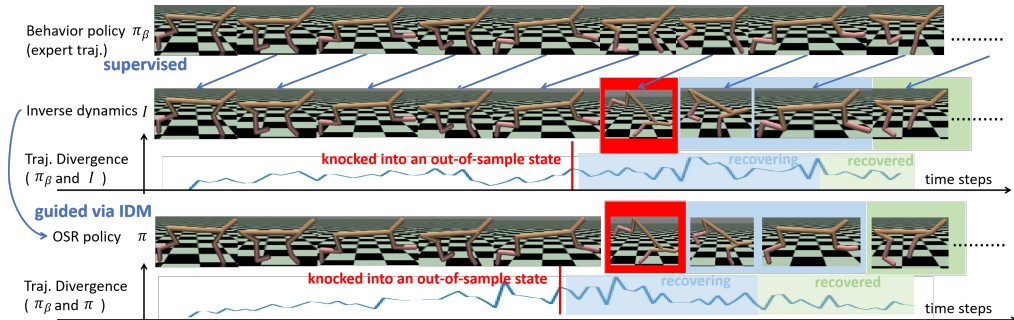

Figure 3: The state recovery process via IDM and OSR on the 'Halfcheetah-OOS-large' benchmark. The interval between every two images is ten steps.

## 5.3 Limited valuable data setting

In this section, we test our method's feasibility when only limited valuable data is available for training, which is a novel task with the mixture of the expert dataset and random dataset with different ratios [27], aiming to evaluate the robustness of ORL algorithms on the condition that the offline dataset only covers limited valuable state space. In this paper, the proportions of random samples are 0.5, 0.6, 0.7, 0.8 and 0.9 and the ratios of expert samples decline, for Halfcheetah, Hopper and Walker2d. All datasets contain 1,000,000 samples.

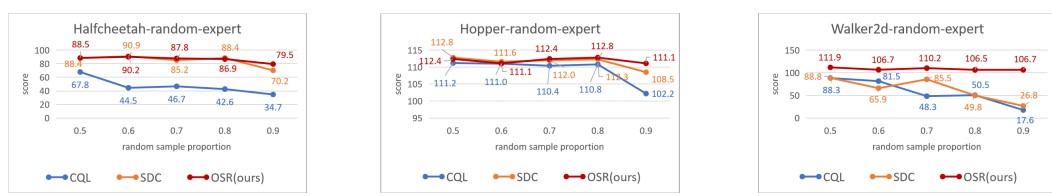

Figure 4: The curve of CQL, SDC and OSR in the setting of limited valuable data.

We compare the proposed OSR with CQL and SDC in a limited valuable data setting. The results, shown in Figure 4, demonstrate that all methods experience performance degradation when expert samples are limited. However, our proposed method, OSR, achieves the lowest performance decrease with the raise of the random sample ratio among the three methods in most tasks, which suggests that our method is less sensitive to the quality of training data. The results above mean that even if the dataset covers only limited valuable area of the extensive state space, the proposed OSR still works.

## 6 Conclusion

In this paper, we proposed a simple yet effective method, named Out-of-sample State Recovery (OSR), for offline reinforcement learning. The proposed method follows the *state recovery* principle to deal with the *state distributional shift* issue commonly encountered in offline RL. Unlike previous methods, we address this in a way that does not need to explicitly model the transition of the underlying environment by treating the inverse dynamics model as a guidance with desired immediate consequence for the new policy. Our theoretical results show that the proposed OSR is equivalent to aligning the transition distribution of the learnt policy with the offline dataset. The proposed method is evaluated on several offline AntMaze/MuJoCo settings and achieved the SOTA performance.

## Acknowledgement

This work is partially supported by National Key R&D program of China (2021ZD0113203), National Science Foundation of China (61976115).

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
