$$\forall \epsilon, \exists \delta, s.t. \|s - \hat{s}\| \leq \epsilon \Rightarrow$$
$$\|P(s'|s,\pi_\beta) - P(s'|\hat{s},\pi_\beta)\| \leq \delta$$

*and $\forall s, s'$, we have $P(s'|s,\pi_\beta) > 0$.*

By Assumption 1, we could easily infer:

$$\forall \epsilon, \exists \delta, s.t. \|s - \hat{s}\| \leq \epsilon \Rightarrow$$
$$1 - \delta \leq \frac{P(s'|s,\pi_\beta)}{P(s'|\hat{s},\pi_\beta)} \leq 1 + \delta$$

**Assumption 2.** *Assume the new policy we train is a soft exploration policy, which choose every action a from the action set A with probability $\epsilon_\pi$ at least at each state s in S, i.e., $\forall s \in S, a \in A, \pi(a|s) \geq \epsilon_\pi > 0$.*

The rationality of this comes from the boundedness of action space $A$. For example, one typical way we model the policy $\pi$ is to construct a Gaussian distribution. Noting the upper bound of the action space $A$ is $a_{upp}$ and the lower bound is $a_{low}$, then $\epsilon_\pi = \min(\pi(a_{upp}|s), \pi(a_{low}|s)) > 0$.

## B  Proofs

### B.1  Proof of Theorem 1

**Theorem 1.** Given two consecutive state $s$ and $s'$, the out-of-sample state $\hat{s}$ within the $\epsilon$-neighbourhood of $s$ and the behavior policy $\pi_\beta$. Given an inverse dynamics model $I^{\pi_\beta}(a|s,s')$. In bounded action space setting, the following two optimization formulas are equivalent,

$$\min_\pi KL\left(\mathbb{E}_{s'\sim P(s'|s,\pi_\beta)} I^{\pi_\beta}(a|\hat{s},s') \middle\| \pi(a|\hat{s})\right) \tag{13}$$

$$\iff \min_\pi KL\left(P(s'|s,\pi_\beta) \middle\| P(s'|\hat{s},\pi)\right) \tag{14}$$

, that is, the two optimization formulas achieve the same policy.

*Proof.*

$$\min_{\pi} KL\left(\mathbb{E}_{s'\sim P(s'|s,\pi_\beta)} I^{\pi_\beta}(a|\hat{s},s')\Big\| \pi(a|\hat{s})\right)$$

$$\Leftrightarrow \min_{\pi} \sum_a \sum_{s'} P(s'|s,\pi_\beta)\frac{\pi_\beta(a|\hat{s})P(s'|\hat{s},a)}{P(s'|\hat{s},\pi_\beta)} \log \sum_{\tilde{s}'} P(s'|s,\pi_\beta)\frac{\pi_\beta(a|\hat{s})P(\tilde{s}'|\hat{s},a)}{P(\tilde{s}'|\hat{s},\pi_\beta)\pi(a|\hat{s})}$$

$$\Leftrightarrow \min_{\pi} \sum_a \sum_{s'} P(s'|s,\pi_\beta)\frac{\pi_\beta(a|\hat{s})P(s'|\hat{s},a)}{P(s'|\hat{s},\pi_\beta)} \log \frac{1}{\pi(a|\hat{s})}$$

$$\leq \min_{\pi} (1+\delta) \sum_a \sum_{s'} \pi_\beta(a|\hat{s})P(s'|\hat{s},a) \log \frac{1}{\pi(a|\hat{s})} \qquad (Assumption1)$$

$$\Leftrightarrow \min_{\pi} \sum_a \sum_{s'} \pi_\beta(a|\hat{s})P(s'|\hat{s},a) \log \frac{\epsilon_\pi P(s'|\hat{s},\pi_\beta)}{\pi(a|\hat{s})}$$

$$\leq \min_{\pi} \sum_a \sum_{s'} \pi_\beta(a|\hat{s})P(s'|\hat{s},a) \log \frac{P(s'|\hat{s},\pi_\beta)}{\sum_a \pi(a|\hat{s})P(s'|\hat{s},a)} \qquad (Assumption2)$$

$$\Leftrightarrow \min_{\pi} \sum_{s'} P(s'|\hat{s},\pi_\beta) \log \frac{P(s'|\hat{s},\pi_\beta)}{P(s'|\hat{s},\pi)}$$

$$\leq \min_{\pi} \sum_{s'} \frac{P(s'|s,\pi_\beta)}{1-\delta} \log \frac{\frac{P(s'|s,\pi_\beta)}{1-\delta}}{P(s'|\hat{s},\pi)} \qquad (Assumption1)$$

$$\Leftrightarrow \min_{\pi} \sum_{s'} P(s'|s,\pi_\beta) \log \frac{P(s'|s,\pi_\beta)}{P(s'|\hat{s},\pi)}$$

$$\Leftrightarrow \min_{\pi} KL\left(P(s'|s,\pi_\beta)\Big\| P(s'|\hat{s},\pi)\right) \qquad (15)$$

On the other hand:

$$\min_{\pi} KL\left(\mathbb{E}_{s'\sim P(s'|s,\pi_\beta)} I^{\pi_\beta}(a|\hat{s},s')\Big\| \pi(a|\hat{s})\right)$$

$$\Leftrightarrow \min_{\pi} \sum_a \sum_{s'} P(s'|s,\pi_\beta)\frac{\pi_\beta(a|\hat{s})P(s'|\hat{s},a)}{P(s'|\hat{s},\pi_\beta)} \log \frac{1}{\pi(a|\hat{s})}$$

$$\geq \min_{\pi} (1-\delta) \sum_a \sum_{s'} \pi_\beta(a|\hat{s})P(s'|\hat{s},a) \log \frac{1}{\sum_a \pi(a|\hat{s})P(s'|\hat{s},a)} \qquad (Assumption1)$$

$$\Leftrightarrow \min_{\pi} \sum_{s'} P(s'|\hat{s},\pi) \log \frac{P(s'|\hat{s},\pi_\beta)}{P(s'|\hat{s},\pi)}$$

$$\geq \min_{\pi} \sum_{s'} \frac{P(s'|s,\pi)}{1+\delta} \log \frac{\frac{P(s'|s,\pi_\beta)}{1+\delta}}{P(s'|\hat{s},\pi)} \qquad (Assumption1)$$

$$\Leftrightarrow \min_{\pi} \sum_{s'} P(s'|s,\pi_\beta) \log \frac{P(s'|s,\pi_\beta)}{P(s'|\hat{s},\pi)}$$

$$\Leftrightarrow \min_{\pi} KL\left(P(s'|s,\pi_\beta)\Big\| P(s'|\hat{s},\pi)\right) \qquad (16)$$

Combining (15) and (16), we complete the proof. $\qquad\square$

## B.2  Proof of Proposition 1

**Proposition 1.**  Given an arbitrary state $s$ and its target state $s'$. $Q$ is an approximal Q-value function and $V$ is its value function. $\pi_Q$ is the learnt policy according to $Q$. Let $\pi_Q(a|s)$ is positive correlated to $Q(s,a)$, noted as $\pi_Q(a|s) \propto Q(s,a)$, and assume $\exists a, P(s'|s,a) > 0$, then $P(s'|s,\pi_Q) \propto V(s')$.

*Proof.*

$$\pi_Q(a|s) \propto Q(s,a) \tag{17}$$

$$\Rightarrow \pi_Q(a|s) \cdot \sum_{s'} P(s'|s,a) \propto R(s,a) + \gamma \sum_{s'} P(s'|s,a)V(s') \tag{18}$$

$$\Rightarrow \sum_{s'} \left[ \pi_Q(a|s) \cdot P(s'|s,a) \right] \propto \sum_{s'} P(s'|s,a) \left[ R(s,a) + \gamma V(s') \right] \tag{19}$$

$$\Rightarrow \left[ \pi_Q(a|s) \cdot P(s'|s,a) \right] \propto P(s'|s,a) \left[ R(s,a) + \gamma V(s') \right] \tag{20}$$

$$\Rightarrow \sum_{a} \left[ \pi_Q(a|s) \cdot P(s'|s,a) \right] \propto \sum_{a} P(s'|s,a)R(s,a) + V(s') \cdot \gamma \sum_{a} P(s'|s,a) \tag{21}$$

$$\Rightarrow P(s'|s,\pi_Q) \propto C_1(s,s') + V(s') \cdot C_2(s,s',\gamma) \tag{22}$$

$$\Rightarrow P(s'|s,\pi_Q) \propto V(s') \tag{23}$$

complete the proof. □

Proposition 1 shows that from any state $s$, overestimating the value of a target state $s'$ would enhance the potentiality for the agent to transit to this target $s'$(except the condition that $s'$ is unreachable from $s$, i.e., $\forall a, P(s'|s,a) = 0$). In this paper, the transition we wish the agent to take is from the noisy $\hat{s}$ of $s$ to the $s'$ which is known reachable from $s$. For the reason that $\hat{s}$ is not very far away from $s$, so we can assume $s'$ is reachable from $\hat{s}$, and by Proposition 1, we conclude that $\mathbb{E}_{s' \sim P(s'|s,\pi_\beta)} P(s'|\hat{s},\pi) \propto E_{s' \sim P(s'|s,\pi_\beta)} V(s')$. For the fact that $P(s'|s,\pi_\beta)$ is the transited state distribution conditioned on $s$ that aligned with the offline dataset $\mathcal{D}$, we conclude overestimating of in-sample $s'$ increases the agent's preference to transiting to in-sample states.

### B.3 Proof of Proposition 2

**Proposition 2.** When training on the original dataset $\mathcal{D}$, $L_{sr-v}$ is equivalent to CQL regularization.

*Proof.* The proof is performed on the equivalence of OSR and CQL regularizations given an in-sample state $s$:

$$\min_Q \mathbb{E}_{a \sim \pi(a|s)} Q(s,a) - \mathbb{E}_{s' \sim P(s'|s,\pi_\beta)} \mathbb{E}_{a \sim I^{\pi_\beta}(a|s,s')} Q(s,a)$$

$$\Leftrightarrow \min_Q \mathbb{E}_{a \sim \pi(a|s)} Q(s,a) - \sum_{s'} P(s'|s,\pi_\beta) \sum_{a} \frac{\pi_\beta(a|s)P(s'|s,a)}{P(s'|s,\pi_\beta)} Q(s,a)$$

$$\Leftrightarrow \min_Q \mathbb{E}_{a \sim \pi(a|s)} Q(s,a) - \sum_{a} \pi_\beta(a|s)Q(s,a)$$

$$\Leftrightarrow \min_Q \mathbb{E}_{a \sim \pi(a|s)} Q(s,a) - \mathbb{E}_{a \sim \pi_\beta(a|s)} Q(s,a)$$

$$\Leftrightarrow \quad CQLregularization$$

Complete the proof. □

### B.4 Proof of Theorem 2

**Theorem 2.** Given two consecutive state $s$ and $s'$, the out-of-sample state $\hat{s}$ within the $\epsilon$-neighbourhood of $s$ and the behavior policy $\pi_\beta$. $\pi$ is the new policy. Define $Q(s,a)$ as a approximal Q-value function and $V(s)$ as a approximal value function. Given an inverse dynamics model $I^{\pi_\beta}(a|s,s')$. Then the following two optimization formulas are approximately equivalent:

$$\min_Q E_{\hat{a} \sim \pi(\cdot|\hat{s})} Q(\hat{s},\hat{a}) - \mathbb{E}_{\hat{a}_{tar} \sim I^{\pi_\beta}(\cdot|\hat{s},s')} Q(\hat{s},\hat{a}_{tar}) \tag{24}$$

$$\overset{\sim}{\Leftrightarrow} \min_V \mathbb{E}_{s' \sim P(s'|\hat{s},\pi)} V(s') - \mathbb{E}_{s' \sim P(s'|s,\pi_\beta)} V(s') \tag{25}$$

, that is, they achieve the same policy approximately. Then the difference of the two optimization formulas is no larger than $2\delta \sum_a \pi_\beta(a|\hat{s})Q(\hat{s},a)$.

*Proof.* First we derive the upper bound of (24):

$$\min_Q E_{\hat{a}\sim\pi(\cdot|\hat{s})}Q(\hat{s},\hat{a}) - \mathbb{E}_{\hat{a}_{tar}\sim I^{\pi_\beta}(\cdot|\hat{s},s')}Q(\hat{s},\hat{a}_{tar})$$

$$\Leftrightarrow \min_Q \sum_a \pi(a|\hat{s})Q(\hat{s},a) - \sum_{s'}P(s'|s,\pi_\beta)\sum_a \frac{\pi_\beta(a|\hat{s})P(s'|\hat{s},a)}{P(s'|\hat{s},\pi_\beta)}Q(\hat{s},a)$$

$$\Leftrightarrow \min_Q \sum_a \left[\pi(a|\hat{s}) - \sum_{s'}P(s'|s,\pi_\beta)\frac{\pi_\beta(a|\hat{s})P(s'|\hat{s},a)}{P(s'|\hat{s},\pi_\beta)}\right]Q(\hat{s},a)$$

$$\leq \min_Q \sum_a \left[\pi(a|\hat{s}) - (1-\delta)\pi_\beta(a|\hat{s})\right]Q(\hat{s},a) \qquad (Assumption\,1)$$

Similarly, we have the lower bound of (24):

$$\min_Q E_{\hat{a}\sim\pi(\cdot|\hat{s})}Q(\hat{s},\hat{a}) - \mathbb{E}_{\hat{a}_{tar}\sim I^{\pi_\beta}(\cdot|\hat{s},s')}Q(\hat{s},\hat{a}_{tar})$$

$$\geq \min_Q \sum_a \left[\pi(a|\hat{s}) - (1+\delta)\pi_\beta(a|\hat{s})\right]Q(\hat{s},a)$$

Then we derive the upper bound of (25):

$$\min_V \mathbb{E}_{s'\sim P(s'|\hat{s},\pi)}V(s') - \mathbb{E}_{s'\sim P(s'|s,\pi_\beta)}V(s')$$

$$\Leftrightarrow \min_V \sum_{s'}P(s'|\hat{s},\pi)V(s') - \sum_{s'}P(s'|s,\pi_\beta)V(s')$$

$$\leq \min_V \sum_{s'}P(s'|\hat{s},\pi)V(s') - \sum_{s'}(1-\delta)P(s'|\hat{s},\pi_\beta)V(s') \qquad (Assumption\,1)$$

$$\Leftrightarrow \min_V \sum_{s'}\sum_a \pi(a|\hat{s})P(s'|\hat{s},a)V(s') - \sum_{s'}(1-\delta)\sum_a \pi_\beta(a|\hat{s})P(s'|\hat{s},a)V(s')$$

$$\Leftrightarrow \min_V \sum_a \left[\pi(a|\hat{s}) - (1-\delta)\pi_\beta(a|\hat{s})\right]\cdot\left[\gamma P(s'|\hat{s},a)V(s') + R(\hat{s},a) - R(\hat{s},a)\right] \qquad (Bellman\,Equation)$$

$$\Leftrightarrow \min_Q \sum_a \left[\pi(a|\hat{s}) - (1-\delta)\pi_\beta(a|\hat{s})\right]\left[Q(\hat{s},a) - R(\hat{s},a)\right]$$

$$\Leftrightarrow \min_Q \sum_a \left[\pi(a|\hat{s}) - (1-\delta)\pi_\beta(a|\hat{s})\right]Q(\hat{s},a)$$

Similarly, we have the lower bound of (25):

$$\min_V \mathbb{E}_{s'\sim P(s'|\hat{s},\pi)}V(s') - \mathbb{E}_{s'\sim P(s'|s,\pi_\beta)}V(s')$$

$$\geq \min_Q \sum_a \left[\pi(a|\hat{s}) - (1+\delta)\pi_\beta(a|\hat{s})\right]Q(\hat{s},a)$$

Therefore, for the face that (24) and (25) have the same upper and lower bounds, we conclude that

$$\min_Q E_{\hat{a}\sim\pi(\cdot|\hat{s})}Q(\hat{s},\hat{a}) - \mathbb{E}_{\hat{a}_{tar}\sim I^{\pi_\beta}(\cdot|\hat{s},s')}Q(\hat{s},\hat{a}_{tar}) \qquad (26)$$

is approximately equivalent to:

$$\min_V \mathbb{E}_{s'\sim P(s'|\hat{s},\pi)}V(s') - \mathbb{E}_{s'\sim P(s'|s,\pi_\beta)}V(s') \qquad (27)$$

for the fact that they would achieve the similar policy. Then we subtract the upper bound $\sum_a \left[\pi(a|\hat{s}) - (1-\delta)\pi_\beta(a|\hat{s})\right]Q(\hat{s},a)$ with the lower bound $\sum_a \left[\pi(a|\hat{s}) - (1+\delta)\pi_\beta(a|\hat{s})\right]Q(\hat{s},a)$, and we can get the bound of the difference between (24) and (25), which is $2\delta\sum_a \pi_\beta(a|\hat{s})Q(\hat{s},a)$.

Complete the proof.

$\square$

## B.5 Explanation of the difference between the theory and implementation

First, we add the original $s$ to the tuple $(\tilde{s}, s')$ in the $D_{tot}$ for clearer derivation, as $(s, \tilde{s}, s')$. And it looks like that the proposed optimization objective (Eq.7) is not equivalent to its actual implementation (Eq.11), i.e.,

$$\mathbb{E}_{(s,\tilde{s})\sim D_{tot}} D_{KL}\left[\mathbb{E}_{s'\sim P(s'|s,\pi_\beta)} I^{\pi_\beta}(a|\tilde{s}, s') \Big\| \pi(a|\tilde{s})\right] \neq \mathbb{E}_{(s,\tilde{s})\sim D_{tot}} \mathbb{E}_{s'\sim P(s'|s,\pi_\beta)} D_{KL}\left[I^{\pi_\beta}(a|\tilde{s}, s') \Big\| \pi(a|\tilde{s})\right]$$

, where $P(s'|s, \pi_\beta)$ is the transition distribution.

However, these two optimization problems are actually equivalent, in the sense that they induce the same solution,

$$\arg\min_\pi \mathbb{E}_{(s,\tilde{s})\sim D_{tot}} D_{KL}\left[\mathbb{E}_{s'\sim P(s'|s,\pi_\beta)} I^{\pi_\beta}(a|\tilde{s}, s') \Big\| \pi(a|\tilde{s})\right]$$

$$= \arg\min_\pi \mathbb{E}_{(s,\tilde{s})\sim D_{tot}} \mathbb{E}_{s'\sim P(s'|s,\pi_\beta)} D_{KL}\left[I^{\pi_\beta}(a|\tilde{s}, s') \Big\| \pi(a|\tilde{s})\right]$$

In what below, we give the detailed derivation,

$$\arg\min_\pi \mathbb{E}_{(s,\tilde{s})\sim D_{tot}} D_{KL}\left[\mathbb{E}_{s'\sim P(s'|s,\pi_\beta)} I^{\pi_\beta}(a|\tilde{s}, s') \Big\| \pi(a|\tilde{s})\right]$$

$$= \arg\min_\pi \mathbb{E}_{(s,\tilde{s})\sim D_{tot}} \sum_a \mathbb{E}_{s'\sim P(s'|s,\pi_\beta)} I^{\pi_\beta}(a|\tilde{s}, s') \log \frac{\mathbb{E}_{s''\sim P(s''|s,\pi_\beta)} I^{\pi_\beta}(a|\tilde{s}, s'')}{\pi(a|\tilde{s})}$$

$$= \arg\min_\pi \mathbb{E}_{(s,\tilde{s})\sim D_{tot}} \mathbb{E}_{s'\sim P(s'|s,\pi_\beta)} \sum_a I^{\pi_\beta}(a|\tilde{s}, s') \log \frac{\mathbb{E}_{s''\sim P(s''|s,\pi_\beta)} I^{\pi_\beta}(a|\tilde{s}, s'')}{\pi(a|\tilde{s})}$$

$$= \arg\min_\pi \mathbb{E}_{(s,\tilde{s})\sim D_{tot}} \mathbb{E}_{s'\sim P(s'|s,\pi_\beta)} \sum_a I^{\pi_\beta}(a|\tilde{s}, s') \log \mathbb{E}_{s''\sim P(s''|s,\pi_\beta)} I^{\pi_\beta}(a|\tilde{s}, s'')$$

$$+ \mathbb{E}_{(s,\tilde{s})\sim D_{tot}} \mathbb{E}_{s'\sim P(s'|s,\pi_\beta)} \sum_a I^{\pi_\beta}(a|\tilde{s}, s') \log \frac{1}{\pi(a|\tilde{s})} \tag{28}$$

Note the term $\mathbb{E}_{(s,\tilde{s})\sim D_{tot}} \mathbb{E}_{s'\sim P(s'|s,\pi_\beta)} \sum_a I^{\pi_\beta}(a|\tilde{s}, s') \log \mathbb{E}_{s''\sim P(s''|s,\pi_\beta)} I^{\pi_\beta}(a|\tilde{s}, s'')$ in Eq.(28) is a constant w.r.t. $\pi$, hence we can remove it as follows,

(28)

$$= \arg\min_\pi \mathbb{E}_{(s,\tilde{s})\sim D_{tot}} \mathbb{E}_{s'\sim P(s'|s,\pi_\beta)} \sum_a I^{\pi_\beta}(a|\tilde{s}, s') \log \frac{1}{\pi(a|\tilde{s})} \tag{29}$$

We remark that $\mathbb{E}_{(s,\tilde{s})\sim D_{tot}} \mathbb{E}_{s'\sim P(s'|s,\pi_\beta)} \sum_a I^{\pi_\beta}(a|\tilde{s}, s') \log I^{\pi_\beta}(a|\tilde{s}, s')$ is a constant w.r.t. $\pi$, so we can add it onto Eq.(29) as follows,

(29)

$$= \arg\min_\pi \mathbb{E}_{(s,\tilde{s})\sim D_{tot}} \mathbb{E}_{s'\sim P(s'|s,\pi_\beta)} \sum_a I^{\pi_\beta}(a|\tilde{s}, s') \log \frac{1}{\pi(a|\tilde{s})}$$

$$+ \mathbb{E}_{(s,\tilde{s})\sim D_{tot}} \mathbb{E}_{s'\sim P(s'|s,\pi_\beta)} \sum_a I^{\pi_\beta}(a|\tilde{s}, s') \log I^{\pi_\beta}(a|\tilde{s}, s')$$

$$= \arg\min_\pi \mathbb{E}_{(s,\tilde{s})\sim D_{tot}} \mathbb{E}_{s'\sim P(s'|s,\pi_\beta)} \sum_a I^{\pi_\beta}(a|\tilde{s}, s') \log \frac{I^{\pi_\beta}(a|\tilde{s}, s')}{\pi(a|\tilde{s})}$$

$$= \arg\min_\pi \mathbb{E}_{(s,\tilde{s})\sim D_{tot}} \mathbb{E}_{s'\sim P(s'|s,\pi_\beta)} D_{KL}\left[I^{\pi_\beta}(a|\tilde{s}, s') \Big\| \pi(a|\tilde{s})\right] \tag{30}$$

And then we can remove the expectation w.r.t. $s'$ in Eq.(30) with Monte Calro approximation with the sample number $N$ as 1,

$$
\arg\min_{\pi} \mathbb{E}_{(s,\tilde{s})\sim D_{tot}} \mathbb{E}_{s'\sim P(s'|s,\pi_\beta)} D_{KL}\left[I^{\pi_\beta}(a|\tilde{s},s') \middle\| \pi(a|\tilde{s})\right]
$$

$$
\approx \arg\min_{\pi} \mathbb{E}_{\tilde{s}\sim D_{tot}} \frac{1}{N}\sum_{i=1}^{N} D_{KL}\left[I^{\pi_\beta}(a|\tilde{s},s'_i) \middle\| \pi(a|\tilde{s})\right]
$$

$$
\approx \arg\min_{\pi} \mathbb{E}_{\tilde{s}\sim D_{tot}} D_{KL}\left[I^{\pi_\beta}(a|\tilde{s},s') \middle\| \pi(a|\tilde{s})\right] \tag{31}
$$

where Eq.(31) is the Eq.(11) in our paper.

## C Alogrithms

The pseudoccode of OSR is listed in Algorithm 1, and the pseudoccode of OSR-v is listed in Algorithm 2.

---

**Algorithm 1** Policy-constraint Out-of-sample State Recovery (OSR)

---

**Input**: mixed offline dataset $\mathcal{D}_{tot}$, maximal update iterations $T$,
**Parameter**: policy network $\pi$, Q-networks $Q_1, Q_2$, inverse dynamics model $I$,
**Output**: learnt policy network $\pi$

1: Initialize the policy network, Q-networks and the inverse dynamics model.
2: Let $t = 0$.
3: **while** $t < T$ **do**
4:   Sample mini-batch of N samples $(\tilde{s}, a, r, s')$ from $\mathcal{D}_{tot}$.
5:   Train the inverse dynamics model $I$.
6:   Feed $\tilde{s}$ to the policy network $\pi$ and get $\mu_\pi, \sigma_\pi$.
7:   Get $\hat{a} = \mu_\pi + \sigma_\pi \cdot \epsilon$, where $\epsilon \sim \mathcal{N}(0,1)$.
8:   Feed $\tilde{s}$ and $s'$ to the inverse dynamics model $I$ and get $\mu_I, \sigma_I$.
9:   Update the policy network $\pi$ according to,

$$
L_{sr} = \mathbb{E}_{\tilde{s},s'\sim\mathcal{D}_{tot}}\left[\log\frac{|\sigma_I|}{|\sigma_\pi|} + tr(\sigma_\pi^{-1}\sigma_I) + (\mu_I - \mu_\pi)^T\sigma_\pi^{-1}(\mu_I - \mu_\pi)\right]
$$

$$
L_\pi = \mathbb{E}_{\tilde{s},s'\sim\mathcal{D}_{tot},\hat{a}\sim\pi(\cdot|\tilde{s})}\left[Q(\tilde{s},\hat{a})\right] + \lambda L_{sr}.
$$

10:   Update the Q-networks according to,

$$
L_Q = E_{s,a,r,s'\sim\mathcal{D}}\left[\alpha \cdot \left(\mathbb{E}_{\hat{a}\sim\pi(\hat{a}|s)}Q(s,\hat{a}) - Q(s,a)\right) + \left(r + \gamma\mathbb{E}_{a'\sim\pi(\cdot|s')}Q(s',a') - Q(s,a)\right)^2\right].
$$

11: **end while**
12: **return** learnt policy network $\pi$.

---

**Algorithm 2** Value-constraint Out-of-sample State Recovery (OSR-v)

---

**Input**: offline dataset $\mathcal{D}_{tot}$, maximal update iterations $T$,
**Parameter**: policy network $\pi$, Q-networks $Q_1, Q_2$, inverse dynamics model $I$,
**Output**: learnt policy network $\pi$

1: Initialize the policy network, Q-networks and the inverse dynamics model.
2: Let $t = 0$.
3: **while** $t < T$ **do**
4:    Sample mini-batch of N samples $(\tilde{s}, a, r, s')$ from $\mathcal{D}_{tot}$.
5:    Train the inverse dynamics model $I$.
6:    Feed $\tilde{s}$ to the policy network $\pi$ and get $\hat{a}$.
7:    Feed $\tilde{s}$ and $s'$ to the inverse dynamics model $I$ and get $\hat{a}_{tar}$.
8:    Update the policy network $\pi$ according to,

$$L_\pi = \mathbb{E}_{\tilde{s} \sim \mathcal{D}_{tot}, \hat{a} \sim \pi(\cdot|\tilde{s})} \big[ Q(\tilde{s}, \hat{a}) \big]$$

9:    Update the Q-networks according to,

$$L_{sr-v} = \mathbb{E}_{\tilde{s}, s' \sim \mathcal{D}_{tot}} \Bigg( \mathbb{E}_{\hat{a} \sim \pi(\cdot|\tilde{s})} Q(\tilde{s}, \hat{a}) - \mathbb{E}_{\hat{a}_{tar} \sim I^{\pi_\beta}(\cdot|\tilde{s}, s')} Q(\tilde{s}, \hat{a}_{tar}) \Bigg)$$

$$L_Q = \alpha \cdot L_{sr-v} + \mathbb{E}_{s,a,r,s' \sim \mathcal{D}} \bigg[ r + \gamma \max_{a'} Q(s', a') - Q(s, a) \bigg]$$

10: **end while**
11: **return** learnt policy network $\pi$.

---

# D   More experimental results

## D.1   Sensitive analysis over hyperparameters

In this section, we perform sensitive analysis on two key hyperparameters: the balance-coefficient $\lambda$ and the noise magnitude $\beta$ to evaluate how these hyperparameters influence the performance of OSR.

**Sensitive analysis on $\beta$.**  The $\beta$ is the hyperparameter that control the magnitude of the noise that add to our mixed dataset $\mathcal{D}_{tot}$ for training. Its influence to OSR is as shown in Figure 5, where the curve is smoothed by a Gaussian filter. The performance of the learnt policy is best when the $\beta$ is 1e-3; takes second place when the $\beta$ equals to 5e-3; is worst when the $\beta$ is 1e-2. From the results, we conclude that when $\beta$ is too large, the agent would fail to be well trained over some or all seeds. Therefore, we ought to conservatively choose the value of $\beta$ and experience suggests the $\beta$ should be no bigger than 2e-3 for 'walker2d' benchmark, 5e-3 for 'hopper' and '3e-3' for 'halfcheetah'.

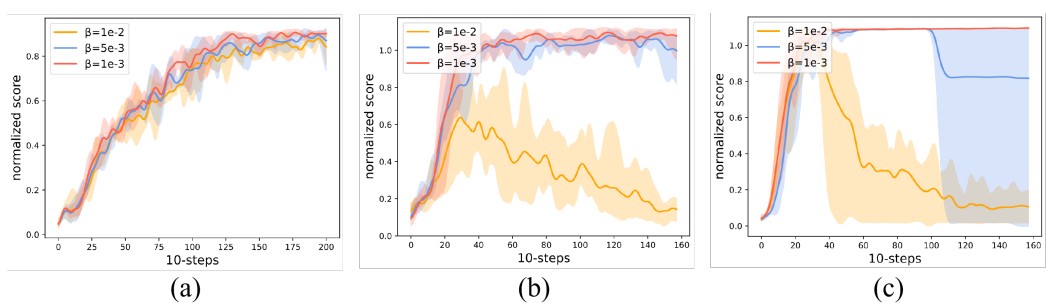

Figure 5: The sensitive results of $\beta$. (a) is on 'halfcheetah-medium-expert', (b) is on 'hopper-medium-expert' and (c) is on 'walker2d-medium-expert'.

**Sensitive analysis on $\lambda$.**  The $\lambda$ is the weight for the OSR regularization that affects the magnitude of conservatism of OSR. The sensitivity of this hyperparameter to the performance of OSR is as

shown in Figure 6, where the curve is smoothed by a Gaussian filter. The learnt policy performs best when $\lambda$ equals to 1.0; slight poor when $\lambda$ is 0.5; worst when $\lambda$ is 0.1. Therefore, we can conclude that if $\lambda$ is too small, OSR may fail to train the policy, and experience suggests that the $\lambda$ should range from 0.5 to 1.0.

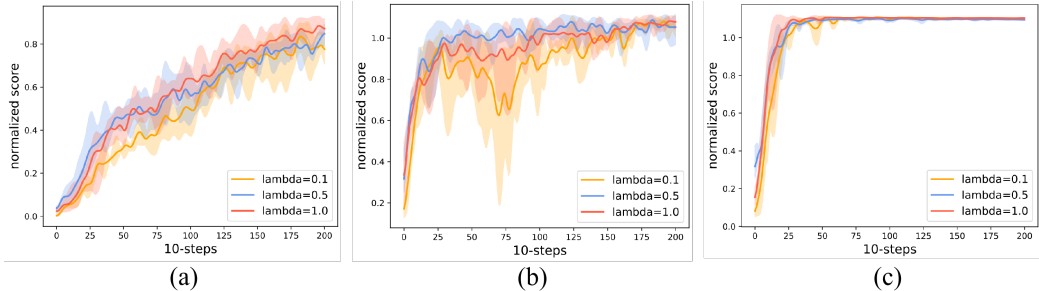

Figure 6: The sensitive results of $\lambda$. (a) is on 'halfcheetah-medium-expert', (b) is on 'hopper-medium-expert' and (c) is on 'walker2d-medium-expert'.

To further evaluate how the hyperparameters $\lambda$ and $\beta$ affect the performance, we have attached more results of sensitive analysis based on the normalized score metrics as follows,

Table 3: Sensitive analysis on 'Halfcheetah-medium-expert'

| $\lambda$ \ $\beta$ | 1e-5 | 1e-4 | 1e-3 | 1e-2 | 1e-1 |
|---|---|---|---|---|---|
| 0(CQL) | 62.4 | 62.4 | 62.4 | 62.4 | 62.4 |
| 0.01 | 58.7 | 59.3 | 64.6 | 62.1 | 46.5 |
| 0.1 | 54.2 | 76.4 | 83.4 | 63.7 | 50.3 |
| 0.5 | 75.4 | 87.9 | **94.6** | 82.3 | 33.6 |
| 1.0 | 73.2 | 89.2 | 92.4 | 46.7 | 34.4 |
| 10.0 | 64.7 | 67.9 | 76.5 | 43.9 | 32.7 |

Table 4: Sensitive analysis on 'Hopper-medium-expert'

| $\lambda$ \ $\beta$ | 1e-5 | 1e-4 | 1e-3 | 1e-2 | 1e-1 |
|---|---|---|---|---|---|
| 0(CQL) | 111.0 | 111.0 | 111.0 | 111.0 | 111.0 |
| 0.01 | 109.3 | 110.2 | 111.6 | 46.3 | 20.6 |
| 0.1 | 111.4 | 112.1 | 111.3 | 29.1 | 18.7 |
| 0.5 | 111.5 | 112.3 | **112.9** | 17.1 | 20.4 |
| 1.0 | 112.1 | 111.7 | **113.0** | 17.4 | 14.5 |
| 10.0 | 98.3 | 70.8 | 69.6 | 22.6 | 13.3 |

Table 5: Sensitive analysis on 'Walker2d-medium-expert'

| $\lambda$ \ $\beta$ | 1e-5 | 1e-4 | 1e-3 | 1e-2 | 1e-1 |
|---|---|---|---|---|---|
| 0(CQL) | 98.7 | 98.7 | 98.7 | 98.7 | 98.7 |
| 0.01 | 101.1 | 102.5 | 104.6 | 94.1 | 33.3 |
| 0.1 | 103.2 | 108.9 | 112.4 | 97.6 | 16.6 |
| 0.5 | 109.4 | 112.6 | **114.1** | 83.3 | 16.8 |
| 1.0 | 108.2 | 110.1 | **113.8** | 84.7 | 20.1 |
| 10.0 | 101.7 | 100.8 | 89.1 | 72.8 | 17.1 |

From the results, we observe that we should be cautious in choosing a $\beta$ that is not very large; otherwise, it could lead to failure. We remark that it is better to choose the appropriate hyperparameters in the neighborhood of the bold data listed in the table above.

## D.2  Ablation study

### D.2.1  Ablation study on the CQL term

We have conducted a more comprehensive ablation study on three environments, shown in the Table below, where each row gives the normalized scores of a specific compared ablated method.

Table 6: Ablation study on the CQL term

|  | Halfcheetah-m.-e. | Hopper-m.-e. | Walker2d-m.-e. |
|---|---|---|---|
| QL | 9.8 | 0.3 | 0.2 |
| QL+BC | 41.2 | 44.7 | 73.6 |
| QL+IDM | 47.5 | 53.9 | 80.2 |
| CQL | 62.4 | 98.7 | 111.0 |
| CQL+BC | 85.7 | 111.8 | 104.3 |
| CQL+IDM(OSR) | **94.7** | **114.3** | **113.1** |

Based on the above results, we have the following observations:

1.the IDM model is useful - we can see that the performance ranking is: OSR > CQL+BC > CQL , where BC denotes traditional behavior cloning (using behavior policy). The IDM can be thought of as a behavior cloning but taking the consequence of an action into consideration when cloning that action.

2.regulating the Q function search space is important for the generalization capability of an offline RL agent - note that if we replace CQL with standard QL in OSR, we have the following performance ranking: OSR>QL+IDM>QL+BC, which shows that the CQL-stlye value function learning is important both for IDM and BC. One possible reason for this is that the agent is trained under the actor-critic framework where both value function and policy play a role, and most importantly, in the setting of offline RL, conservative learning like CQL is critical for suppressing extrapolation error and overestimation, as pointed out by the reviewer.

3.IDM and CQL are complementary to each other - the IDM provides a way to guide the agent to navigate to safer regions, which allows the agent to learn more smart behavior when encountering unfamiliar states - in the sense that the desired behaviors of an agent should be rational (i.e., being less likely to be punished by the objective function of CQL ) not only under normal in-distribution states but under difficult unseen situations as well (this latter point is less studied in current literature) . The IDM can also be thought of as a mechanism to control the training procedure of CQL, such that it will behave less 'overly-conservatively' during learning, potentially improving its generalization capability.

### D.2.2  Ablation study on the noise injection

We have conducted a more comprehensive ablation study, and the results are shown in the Table below,

Table 7: Ablation study on the noise injection

|  | Halfcheetah-m.-e. | Hopper-m.-e. | Walker2d-m.-e. |
|---|---|---|---|
| CQL | 62.4 | 98.7 | 111.0 |
| CQL+BC | 85.7 | 111.8 | 104.3 |
| CQL+BC(s.) | 91.8 | 111.2 | 109.2 |
| CQL+BC(s.a.) | 88.3 | 111.4 | 108.9 |
| OSR($\beta$=0) | 92.3 | 111.8 | 110.1 |
| OSR | **94.7** | **114.3** | **113.1** |

First, we run our OSR algorithm with the noisy level being zero and compare it with CQL - an OSR baseline without the inverse dynamic model (IDM). The results indicate that our OSR ($\beta$=0) (the 5th row) outperforms CQL (the 1st row) significantly on the 3 tasks. This demonstrates the significance of the inverse dynamics model (IDM) in helping the agent to navigate to more safe regions. Similar performance improvement can be observed if we replace the IDM model with the behavior policy (see the CQL + BC setting, the second row), but its effect is not so significant as our OSR method.

Then we introduce the state noise into our base model OSR ($\beta$=0). The results show that this further improves the performance of the OSR by 2-3 (last row), and adding noise improves the performance of CQL + BC (the 3rd and the 4th row) as well except on the Hopper task, while the proposed OSR(>0) method consistently improves the performance over all the three environments.

## D.3 Training details

In this section, we introduce our training details, including: 1) the hyperparameters and how they were choosen; 2) the structure of the neural networks we use: the Q-networks, inverse dynamics model network and policy network; 3) the total amount of compute and the type of resources used.

### D.3.1 Hyperparameters

In Table 8 and 9, we give the hyperparameters used by OSR and OSR-v to generate Table 1 results. In fact, we have already evaluated the influence of the hyperparameters noise magnitude $\beta$ and SDC weight $\lambda$ in Appendix D.1.

Table 8: The hyperparameters used by OSR to generate Table 1 results.

| Task name | noise magnitude $\beta$ | OSR weight $\lambda$ | CQL weight $\alpha$ |
|---|---|---|---|
| Halfcheetah-random | 1e-3 | 0.5 | 10.0 |
| Walker2d-random | 1e-3 | 0.5 | 5.0 |
| Hopper-random | 1e-3 | 0.5 | 5.0 |
| Halfcheetah-medium | 3e-3 | 0.5 | 10.0 |
| Walker2d-medium | 2e-3 | 0.5 | 5.0 |
| Hopper-medium | 5e-3 | 0.5 | 5.0 |
| Halfcheetah-medium-r. | 3e-3 | 0.5 | 10.0 |
| Walker2d-medium-r. | 2e-3 | 0.5 | 5.0 |
| Hopper-medium-r. | 5e-3 | 0.5 | 5.0 |
| Halfcheetah-medium-e. | 3e-3 | 0.5 | 10.0 |
| Walker2d-medium-e. | 2e-3 | 0.5 | 5.0 |
| Hopper-medium-e. | 5e-3 | 0.5 | 5.0 |
| Halfcheetah-expert | 3e-3 | 0.5 | 10.0 |
| Walker2d-expert | 2e-3 | 0.5 | 5.0 |
| Hopper-expert | 5e-3 | 0.5 | 5.0 |
| AntMaze-umaze | 1e-3 | 0.5 | 5.0 |
| AntMaze-medium-p. | 1e-3 | 0.5 | 5.0 |
| AntMaze-large-p. | 1e-3 | 0.5 | 5.0 |
| AntMaze-umaze-d. | 1e-3 | 0.5 | 5.0 |
| AntMaze-medium-d. | 1e-3 | 0.5 | 5.0 |
| AntMaze-large-d. | 1e-3 | 0.5 | 5.0 |

Table 9: The hyperparameters used by OSR-v to generate Table 1 results.

| Task name | noise magnitude $\beta$ | CQL-OSR weight $\alpha$ |
|---|---|---|
| Halfcheetah-random | 1e-3 | 10.0 |
| Walker2d-random | 1e-3 | 5.0 |
| Hopper-random | 1e-3 | 5.0 |
| Halfcheetah-medium | 3e-3 | 10.0 |
| Walker2d-medium | 2e-3 | 5.0 |
| Hopper-medium | 5e-3 | 5.0 |
| Halfcheetah-medium-r. | 3e-3 | 10.0 |
| Walker2d-medium-r. | 2e-3 | 5.0 |
| Hopper-medium-r. | 5e-3 | 5.0 |
| Halfcheetah-medium-e. | 3e-3 | 10.0 |
| Walker2d-medium-e. | 2e-3 | 5.0 |
| Hopper-medium-e. | 5e-3 | 5.0 |
| Halfcheetah-expert | 3e-3 | 10.0 |
| Walker2d-expert | 2e-3 | 5.0 |
| Hopper-expert | 5e-3 | 5.0 |
| AntMaze-umaze | 1e-3 | 5.0 |
| AntMaze-medium-p. | 1e-3 | 5.0 |
| AntMaze-large-p. | 1e-3 | 5.0 |
| AntMaze-umaze-d. | 1e-3 | 5.0 |
| AntMaze-medium-d. | 1e-3 | 5.0 |
| AntMaze-large-d. | 1e-3 | 5.0 |

### D.3.2 Neural network structures

In this section, we introduce the structure of the networks we use in this paper: policy network, Q network and the inverse dynamics model network.

The structure of the policy network and Q networks is as shown in Table 10, where 's_dim' is the dimension of states and 'a_dim' is the dimension of actions. 'h_dim' is the dimension of the hidden layers, which is usually 256 in our experiments. The policy network is a Guassian policy and the Q networks includes two Q function networks and two target Q function networks.

Table 10: The structure of the policy net and the Q networks.

| policy net | Q net |
|---|---|
| Linear(s_dim, 256) | Linear(s_dim, h_dim) |
| Relu() | Relu() |
| Linear(h_dim, h_dim) | Linear(h_dim, h_dim) |
| Relu() | Relu() |
| Linear(h_dim, a_dim) | Linear(h_dim, 1) |

The structure of the inverse dynamics network is as shown in Table 11, which is a conditional variational auto-encoder. 's_dim' is the dimension of states, 'a_dim' is the dimension of actions and 'h_dim' is the dimension of the hidden variables. 'z_dim' is the dimension of the Gaussian hidden variables in conditional variational auto-encoder.

### D.3.3 Compute resources

We conducted our experiments using a server equipped with one Intel Xeon Gold 6226R CPU, with 16 cores and 32 threads, and 128GB of DDR4 memory. We used a NVIDIA RTX3090 GPU with 24GB of memory for our deep learning experiments. All computations were performed using Python 3.7 and the PyTorch 1.8.1 deep learning framework.

Table 11: The structure of the inverse dynamics model network.

| inverse dynamics model net |
| --- |
| Linear(2 * s_dim, h_dim) |
| Linear(h_dim, h_dim) |
| Linear(h_dim, h_dim) |
| Linear(h_dim, z_dim) | Linear(h_dim, z_dim) |
| Linear(2 * s_dim + z_dim, h_dim) |
| Linear(h_dim, h_dim) |
| Linear(h_dim, a_dim) |

## D.4 Comparison study of methods with Q-ensemble trick

Besides, we also implement OSR with Q-ensemble trick [4], termed as OSR-10, like RORL [25], SAC-10 and EDAC-10 [4], with 10 Q-functions. The results are shown in Figure 12 in Appendix D.4. We observe that the Halfcheetah task improves obviously in SOTA results when using OSR-10, potentially due to its heightened sensitivity to out-of-sample situations.

Table 12: Results of **OSR-10(ours)** and SAC-10, EDAC-10, RORL on 9 MuJoCo benchmarks. Part of the results are reported in the RORL paper. The top-2 highest scores in each benchmark are bolded.

| Methos | Halfcheetah | | | Walker2d | | | Hopper | | |
| --- | --- | --- | --- | --- | --- | --- | --- | --- | --- |
| | medium | medium-r. | medium-e. | medium | medium-r. | medium-e. | medium | medium-r. | medium-e. |
| RORL | **66.8±0.7** | 61.9±1.5 | **107.8±1.1** | **102.4±1.4** | 90.4±0.5 | **121.2±1.5** | **104.8±0.1** | 102.8±0.5 | **112.7±0.2** |
| SAC-10 | 64.9±1.3 | **63.2±0.6** | 107.1±2.0 | 46.7±45.3 | 89.6±3.1 | 116.7±1.9 | 0.8±0.2 | **102.9±0.9** | 6.1±7.7 |
| EDAC-10 | 64.1±1.1 | 60.1±0.3 | 107.2±1.0 | 87.6±11.0 | **94.0±1.2** | 115.4±0.5 | 103.6±0.2 | 102.8±0.3 | 58.1±22.3 |
| OSR-10 | **67.1±0.9** | **64.7±1.7** | **108.7±0.9** | **102.0±0.8** | **93.8±1.3** | **123.4±2.2** | **105.5±0.6** | **103.1±1.2** | 113.2±2.7 |

## D.5 More experimental details on OOSMuJoCo

### D.5.1 State distributional visualized results

To assess the feasibility of our proposed approach, we used t-Distributed Stochastic Neighbour Embedding (t-SNE) [10] to visualize the state distributions of several benchmarks: Halfcheetah-OOS, Hopper-OOS and Walker2d-OOS, as shown in Figure 8, 9 and 7. The results indicate that both SDC and our proposed method, OSR, generate significantly fewer out-of-distribution (OOD) samples, that is, the states generated by SDC and OSR lay in the support of the dataset. In contrast, traditional CQL tends to deviate from the dataset when making decisions at out-of-sample states.

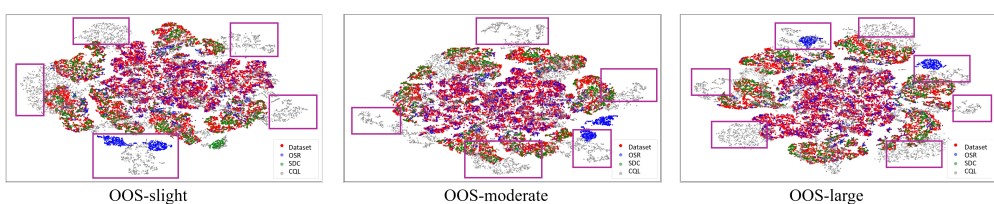

Figure 7: The visualizations of the state distributions of experiments on the 'Halfcheetah-OOS' with three magnitudes. The purple boxes indicate some typical out-of-sample trajectories.

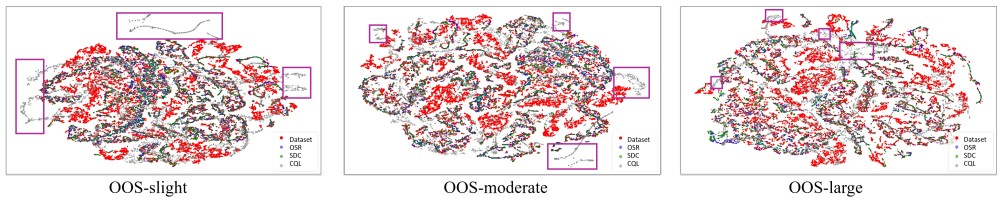

OOS-slight                     OOS-moderate                 OOS-large

Figure 8: The visualizations of the state distributions of experiments on the 'Hopper-OOS' with three magnitudes. The purple boxes indicate some typical out-of-sample trajectories.

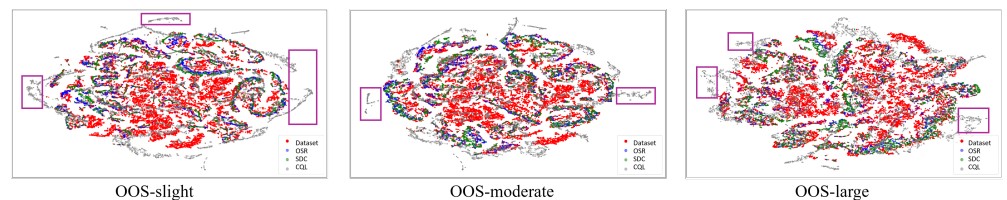

OOS-slight                     OOS-moderate                 OOS-large

Figure 9: The visualizations of the state distributions of experiments on the 'Walker2d-OOS' with three magnitudes. The purple boxes indicate some typical out-of-sample trajectories.

### D.5.2 Other state recovery processes

We also visualize the Hopper and Walker2d agents respectively within an expert trajectory produced by the behavior policy $\pi_\beta$ on the offline dataset $\mathcal{D}$, as is shown in Figure 10 and 11(every 10 steps apart), then we visualize the trajectories by the inverse dynamics model(IDM) guided by the expert trajectory and the new policy of OSR. We add external force to knock the agent into an out-of-sample state and we observe that the agents with IDM and the new policy are all able to recover from the out-of-sample states.

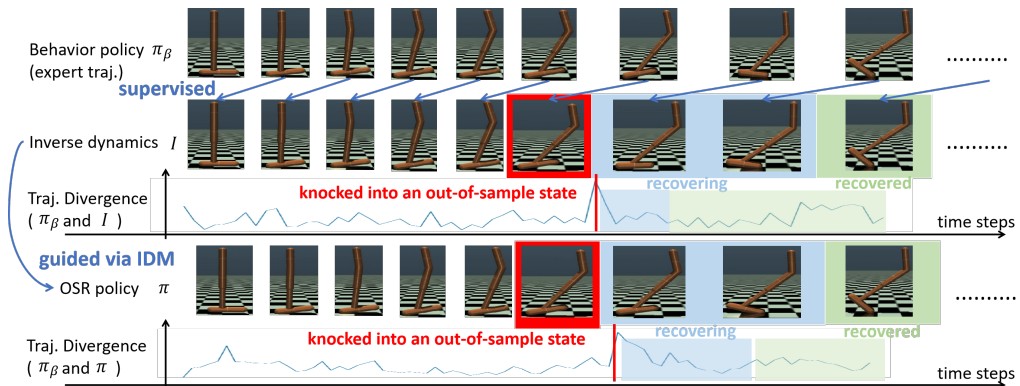

Figure 10: The state recovery process via IDM and OSR on the 'Hopper-OOS-large' benchmark. The interval between every two images is ten steps.

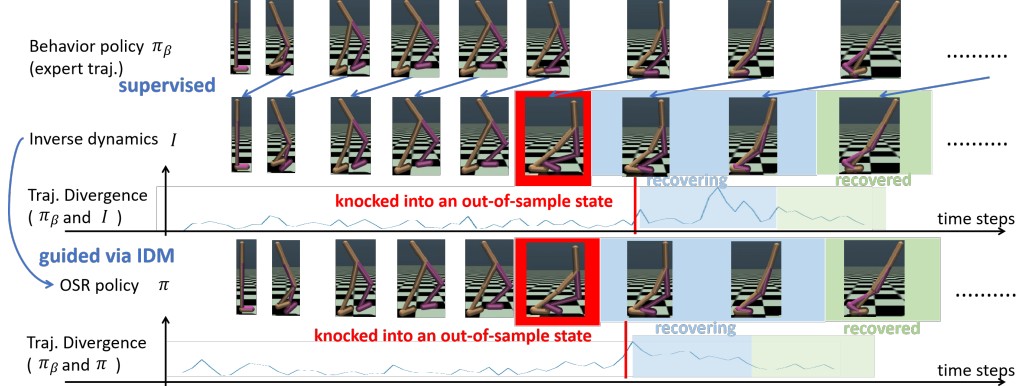

Figure 11: The state recovery process via IDM and OSR on the 'Walker2d-OOS-large' benchmark. The interval between every two images is ten steps.

In order to observe the process of state recovery more clearly, we visualized the motion of each step of the three agents during the state recovery period via the learnt policy of OSR, as is shown in Figure 12. We remark that OSR is able to help the agents to recover from the out-of-sample states, which is quite important for real-world applications.

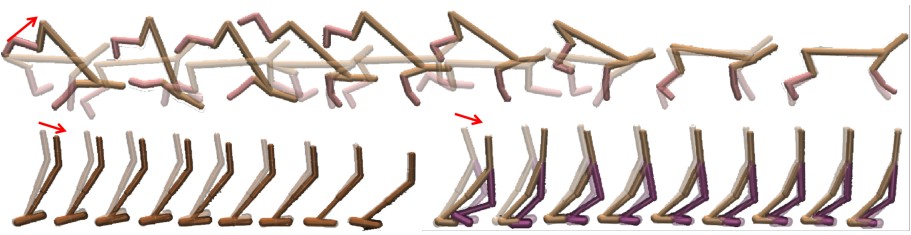

Figure 12: The state recovery processes of three agents: Halfcheetah, Hopper and Walker2d. The red arrows represent the external forces that knock the agents falling into out-of-sample states.

### D.5.3 Comparison with other robust offline RL framework

We have compared our method with RORL [25] on the out-of-sample MuJoCo benchmark, and the results are given below; for comparison, we also give the results obtained without using the out-of-samples on these benchmarks. From the table, we observe that RORL generalizes well at out-of-sample states with slight noise, but its performance may suffer a lot under large perturbation. For example, while its normalized score on Halfcheetah-OOS-s. is 103.8, its performance drops significantly to 55.6 on the more challenging Halfcheetah-OOS-l. benchmark. On the contrary, our method performs much more stable across all the environments. In the attached PDF file, we give some visualization of the trajectories generated by both methods in some OOS benchmarks.

Table 13: Comparison with RORL on the out-of-sample MuJoCo benchmark

|  | RORL-OOS | RORL-w/o OOS | OSR-OOS | OSR-w/o OOS |
|---|---|---|---|---|
| **Halfcheetah-OOS-s.** | 103.8 | 107.8 | 94.1 | 94.7 |
| **Halfcheetah-OOS-m.** | 79.8 | 107.8 | 92.7 | 94.7 |
| **Halfcheetah-OOS-l.** | 55.6 | 107.8 | 91.7 | 94.7 |
| **Hopper-OOS-s.** | 111.5 | 121.2 | 113.3 | 114.3 |
| **Hopper-OOS-m.** | 89.5 | 121.2 | 113.2 | 114.3 |
| **Hopper-OOS-l.** | 66.5 | 121.2 | 110.1 | 114.3 |
| **Walker2d-OOD-s.** | 117.8 | 112.7 | 111.4 | 113.1 |
| **Walker2d-OOS-m.** | 82.2 | 112.7 | 109.2 | 113.1 |
| **Walker2d-OOS-l.** | 46.5 | 112.7 | 106.1 | 113.1 |

## D.6 Comparison with offline RL works with reverse model

we have compared our method with ROMI [23] - an offline RL method based on RDM, and the results are given below. This shows that although, on average, both methods perform comparably across the environments (average score: 68.9 (OSR) vs. 68.2 (ROMI)), our method performs much better in most 'medium' and 'medium-expert' benchmarks (e.g., Halfcheetah-m.-e., Hopper-m., Hopper-m.e., Walker2d-m.e.) than ROMI, indicating the advantage of using IDM if the underlying dataset covering a portion of high-value areas. However, our method may suffer from very noisy datasets (e.g., Hopper-r.), in which case RDM works better. This suggests that it could be beneficial to combine the advantages of both models for an even more robust offline RL.

Table 14: Comparison with ROMI on the standard MuJoCo benchmark

|  | ROMI | OSR(ours) |
|---|---|---|
| **Halfcheetah-m.** | **49.1** | 48.8 |
| **Halfcheetah-m.-r.** | **47.0** | 46.8 |
| **Halfcheetah-m.-e.** | 86.8 | **94.7** |
| **Halfcheetah-r.** | 24.5 | **35.2** |
| **Hopper-m.** | 72.3 | **83.1** |
| **Hopper-m.-r.** | **98.1** | 96.7 |
| **Hopper-m.-e.** | 111.4 | **113.1** |
| **Hopper-r.** | **30.2** | 10.3 |
| **Walker2d-m.** | 84.3 | **85.7** |
| **Walker2d-m.-r.** | **109.7** | 87.9 |
| **Walker2d-m.-e.** | 109.7 | **114.3** |
| **Walker2d-r.** | 7.5 | **13.5** |

## D.7 Comparison on the MuJoCo benchmark with adversarial attacks

We have trained our method OSR-10 based on the large perturbation scales in the following table (same as Table 5 in [25]), referred as OSR-10-large-noise (OSR-10-l.), on the three 'medium' MuJoCo datasets, while OSR-10 and RORL utilize the perturbation with smaller scales metioned before.

Table 15: Hyperparameters of OSR-10-large-noise (OSR-10-l.)

|  | **Halfcheetah-medium** | **Hopper-medium** | **Walker2d-medium** |
|---|---|---|---|
| Scalar $\epsilon_{OOD}$ of OOD Loss | 0.00 | 0.02 | 0.03 |
| Scalar $\epsilon_{OSR}$ of OSR Loss | 0.05 | 0.005 | 0.07 |

We run OSR-10-l. and RORL-l. (RORL trained on the perturbation scales in the Table 5 of [25] on 'medium' as well) in both the adversarial environments introduced in [25] and the OOS MuJoCo benchmarks proposed in our paper. The results of both methods in the adversarial environments are listed in the table below, where we set the scale of perturbation of the adversarial environments as 0.15, and we also provide the two methods' results in the clean environments for comparison (3rd and 6th rows).

The results show that the peformance of RORL and OSR-10 improves by 13.1% and 12.1% respectively afer adding larger scale of perturbation in training. This indicates that increasing the difficulty of training data to some degree is indeed helpful in enhancing the generalization capbility of the learned agent, hence improving its robustness against adversarial attacks. It's our future work to further investigate the performance boundary of applying this trick.

Table 16: Comparison on the MuJoCo benchmark with adversarial attacks with RORL

| | RORL-OOS | RORL-l.-OOS | RORL-l.-w/o OOS | OSR-10-OOS | OSR-10-l.-OOS | OSR-10-l.-w/o OOS |
|---|---|---|---|---|---|---|
| **Halfcheetah+random** | 53.4 | 58.4 | 61.2 | 55.2 | 59.3 | 63.3 |
| **Halfcheetah+action diff** | 51.7 | 52.1 | 61.2 | 53.3 | 54.6 | 63.3 |
| **Halfcheetah+min Q** | 43.9 | 46.4 | 61.2 | 50.6 | 55.0 | 63.3 |
| **Hopper+random** | 67.8 | 82.8 | 102.8 | 70.6 | 87.9 | 103.5 |
| **Hopper+action diff** | 62.3 | 77.3 | 102.8 | 63.8 | 76.2 | 103.5 |
| **Hopper+min Q** | 41.3 | 46.2 | 102.8 | 44.9 | 49.4 | 103.5 |
| **Walker2d+random** | 76.7 | 93.1 | 97.4 | 79.4 | 91.7 | 100.7 |
| **Walker2d+action diff** | 76.4 | 89.2 | 97.4 | 79.1 | 92.8 | 100.7 |
| **Walker2d+min Q** | 68.6 | 72.5 | 97.4 | 78.9 | 81.6 | 100.7 |

## D.8 Experiments on out-of-distribution MuJoCo benchmark

To assess the generalizability of our method, we need to first construct a dataset with more realistic out-of-sample states to test the learnt model. For this we employ a modified generative adversarial network (GAN), which is optimized as follows,

$$\min_{G} \max_{D}[E_{s \sim P(s)}[\log D(s)] + E_{P_G(s')}[\log(1 - D(s'))]] + \alpha \cdot E_{P_{G(s)}} H[\pi_\beta(\cdot|s)] \quad (32)$$

where $G$ is generator, $D$ is the discriminator, $P(s)$ is the real-word state distribution (the dataset) and $P_G(s)$ is the state distribution generated via $G$. $H$ is the entropy function. In words, the above objective aims to generate real-world samples (i..e., consistent with the in-distribution data) that confuse the behavior policy $\pi_\beta$ most; hence its output could be considered as kind of realistic out-of-sample states.

In evaluation, we initialize the MuJoCo environments with the generated out-of-sample states and assess the generalizability of the learnt agent on them. The results, based on the average of normalized scores and recovery rate, are as follows:

Table 17: Normalized scores/recovery rate on the generated out-of-distribution MuJoCo states

| | Halfcheetah | Hopper | Walker2d |
|---|---|---|---|
| CQL | 20.4/33.8% | 36.5/39.1% | 13.1/12.6% |
| OSR | **40.1/69.9%** | **72.0/88.8%** | **43.3/32.4%** |

The above results indicate that our OSR effectively guides the agent to recover from most real-world out-of-sample situations (nearly 70%) in the Halfcheetah and Hopper tasks, despite the mismatch between the constructed (Gaussian) noisy dataset and the actual (GAN-based) out-of-sample states. Although the Walker2D task seems challenging to it, our OSR method performs significantly better than CQL, where a risk-guiding mechanism like ours is lacking.

To gain further insights, we provide visualizations of typical out-of-distribution (OOD) states in the three tasks, along with the corresponding trajectories of OSR, as is shown in Figure 13.

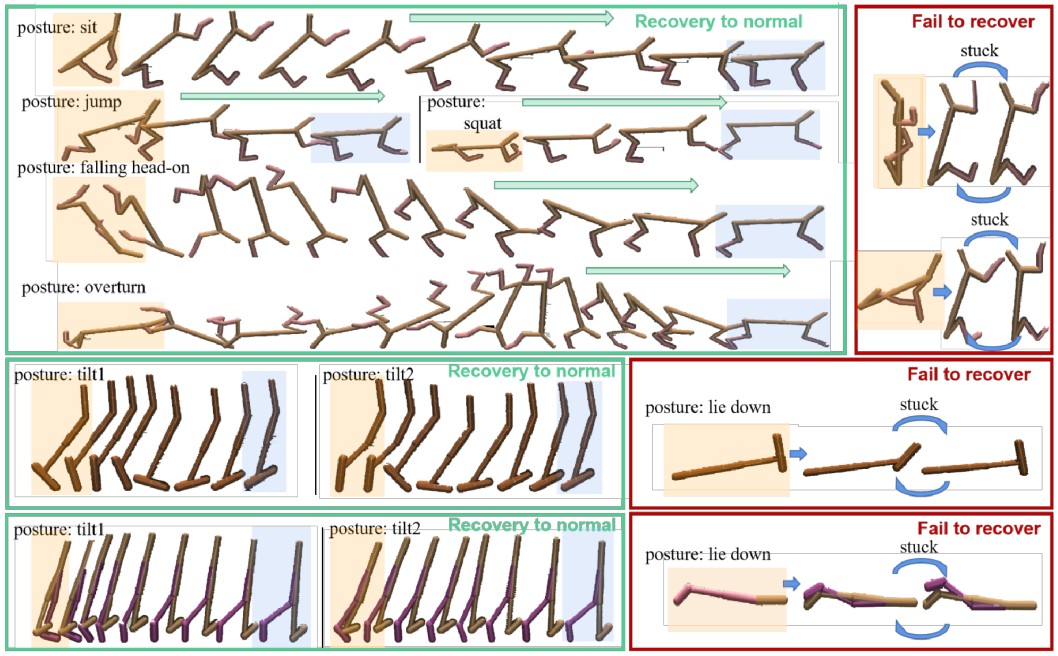

Figure 13: The visualization of OSR's state recovery process on the real-world out-of-sample (OOS) Halfcheetah, Hopper and Walker2d benchmarks, where the initial states are generated by the modified generative adversarial network. The states in orange boxes represent the generated OOS states, and those in blue boxes are the recovered normal states. The trajectories in the left green box are samples successfully recovered while in the right red box are samples failed to recover.

## D.9 Code

We build our OSR and OSR-v based on the CQL project from github[3]. The reasons why we choose YOUNG-GENG's CQL project instead of the official version[4] are as follows: 1) The official version CQL code perform incorrect gradient calculation with old version of pytorch(see issue #5); 2)In the project by yong-geng, a Gaussian policy is utilized, which meets the needs of our method.

We build our OSR-10 based on the RORL project from github[5].

## E  Discussion

**The assumptions.**  The theoretical framework presented in this paper is based on two key assumptions. Assumption 1 assumes that the transition of the behavior policy is 'smooth-transitioned', meaning that it is insensitive to disturbances in observations. However, in practical applications, the behavior policy may not have such a property, which could limit the generalizability of the proposed method to some datasets. Therefore, it is important to carefully consider the effects of non-smooth transitions when applying the proposed method in practice. Assumption 2, on the other hand, assumes that the new policy is modeled as a Gaussian distribution. While this assumption is reasonable in many settings, it may not hold if the new policy is modeled in a different way, such as a deterministic policy, then the validity of Assumption 2 should be carefully evaluated based on the specific modeling assumptions used in each application. Therefore, it is important to note that the limitations of the proposed method are closely tied to the assumptions made in the theoretical framework.

---

[3]Project of CQL by YOUNG-GENG: https://github.com/young-geng/CQL
[4]Project of CQL by AVIRALKUMAR2907: https://github.com/aviralkumar2907/CQL
[5]Project of CQL by YangRui2015: https://github.com/YangRui2015/RORL

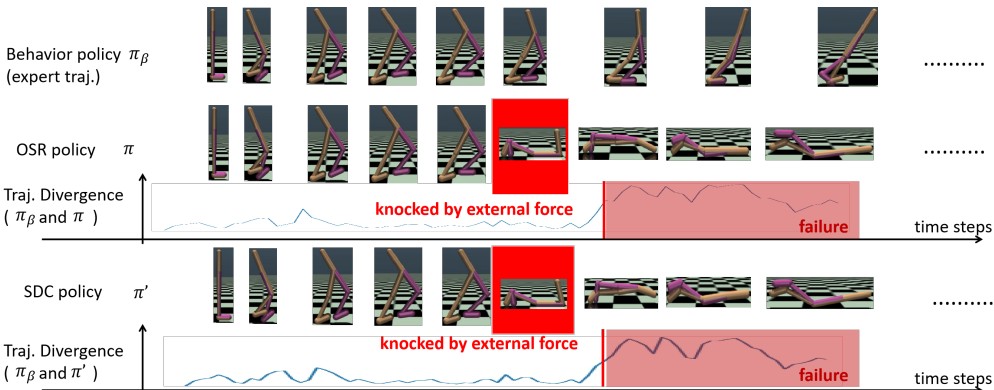

Figure 14: The state recovery process via IDM and OSR on the Walker2d-OOS benchmark with vary large knock(100 steps of random actions). The interval between every two images is ten steps.

**Not all kind of out-of-sample situations could be recovered.** In a related work by [27], the authors identify three key factors that can lead to *state distributional shift* problem in offline reinforcement learning: initial state difference, dynamics bias, and approximation error. Initial state difference refers to the difference between the initial state distributions of the offline dataset and the actual environment, and dynamics bias arises due to the discrepancy between the empirical distribution of the dynamics of the offline dataset and the true dynamics. Approximation error of the neural network can also contribute to *state distributional shift*. However, it is important to note that in scenarios where the initial state difference and dynamics bias drive the agent to a state that is quite far from the dataset, neither SDC nor our proposed method, OSR, can enable the agent to recover to the dataset, for the lack of the information for the inference of such a recovery, as is shown in Figure 14. Then additional strategies may be required to address these challenges. Nonetheless, both SDC and OSR can effectively address the problems of approximation error and some perturbed environments, where the out-of-sample states are located within the $\beta$ neighborhood of the dataset.