# OpenReview forum: "Recovering from Out-of-sample States via Inverse Dynamics in Offline Reinforcement Learning"
_NeurIPS.cc/2023/Conference — NeurIPS 2023 poster_

### Official Review · Reviewer_cN6c · 2023-07-01

**Soundness:** 4 excellent
**Presentation:** 4 excellent
**Contribution:** 3 good
**Rating:** 7
**Confidence:** 4

**Summary:**

This paper aims to tackle a critical challenge in offline reinforcement learning, which involves recovering the state distribution during testing from out-of-sample states. To address this, the authors propose two methods, OSR and OSR-v, which leverage a learned inverse dynamics model to regularize the policy and underestimate the value function. The authors provide theoretical evidence supporting OSR's ability to align the transited state distribution of the new policy with the offline dataset distribution at out-of-sample states. Extensive experiments demonstrate that these proposed methods achieve state-of-the-art performance on general offline RL benchmarks as well as an out-of-sample MuJoCo benchmark.

**Strengths:**

* Overall, the paper is well-written and easy to follow.
* This paper studies a significant problem in offline reinforcement learning, focusing on state distribution correction. The proposed methods show a link between State Deviation Correction (SDC) and robust offline RL.
* The proposed method achieves state-of-the-art (SOTA) performance in both two general offline RL benchmark and an out-of-sample MuJoCo benchmark.
* The effectiveness of OSR in recovering from specific perturbations is validated through visualizations presented in Section 5.2 and Appendix 4.2.
* The code necessary for reproducing the results is provided.

**Weaknesses:**

* Although this paper has already conducted numerous experiments, I believe that a more comprehensive ablation study would be valuable. Specifically, in Appendix D.1, all the figures demonstrate that the smallest $\beta$ achieves the best performance. I am curious to know the performance of OSR when $\beta=0$, meaning the utilization of only $\mathcal{D}$ for training. Theoretical expectations indicate that OSR-v's performance should align with CQL, while OSR's performance should match that of CQL+BC. If OSR with $\beta=0$ still outperforms CQL and CQL+BC, it suggests that the action noise injected by the reverse model plays a crucial role in achieving a smooth policy output. Furthermore, conducting additional comparisons between CQL/CQL+BC with vanilla action noise would provide further insights.
* I have concerns regarding Eq (5), as it assigns the same scale of Gaussian noise to different dimensions. Perhaps normalizing the states by their mean and standard deviation, similar to prior work [1], would be beneficial.
* I think the proposed method also has a relevance to robust RL, as demonstrated by the Out-of-sample MuJoCo experiments which highlight OSR's capability to withstand testing-time attacks. Consequently, it is important to discuss prior works in robust RL and robust offline RL in the related works section, and to provide comparisons with existing robust offline RL approaches in the Out-of-sample MuJoCo benchmark.
* The literature review lacks an inclusion of prior offline RL works such as ROMI [2], which incorporates a reverse model for data augmentation. Hence, the authors are suggested to further engage in a discussion and comparison with ROMI to enhance the comprehensiveness of the review.


[1] Yang R, Bai C, Ma X, et al. Rorl: Robust offline reinforcement learning via conservative smoothing[J]. Advances in Neural Information Processing Systems, 2022, 35: 23851-23866.

[2] Wang J, Li W, Jiang H, et al. Offline reinforcement learning with reverse model-based imagination. Advances in Neural Information Processing Systems, 2021, 34: 29420-29432.

**Questions:**

* Can the authors provide a more comprehensive ablation study, which should include scenarios such as OSR with $\beta=0$, a comparison with CQL+BC, CQL+BC with state noise, and CQL+BC with both state and action noise? This will effectively illustrate the advantages of OSR and demonstrate the significance of incorporating state noise and the inverse model.
* Would normalizing the states by their mean and standard deviation be more useful in addressing perturbations across different state dimensions and ensuring a more consistent $\beta$?
* Additionally, it is recommended to include an additional subsection in the related work that focuses on prior works in robust RL and robust offline RL.
* It is suggested to compare prior robust offline RL work, such as RORL, against the out-of-sample MuJoCo benchmark
* Furthermore, it is necessary to engage in further discussion and conduct comparisons with prior offline RL works that utilize a reverse model.



**Limitations:**

The limitation highlighted by the authors in this work is the challenge of applying it in scenarios where the two basic assumptions, namely the smooth behavior policy transition and Gaussian policy assumption, do not hold.

---

> ### Author Rebuttal · Authors · 2023-08-08
>
> Thanks for your thoughtful comments. We provide clarification to your questions as below. We appreciate it if you have any further questions or comments.
>
> **Q1:... more comprehensive ablation study...**
>
> **Response:** Per your suggestion, we have conducted a more comprehensive ablation study, and the results are shown in the Table below,
>
> | |Halfcheetah-m.-e.|Hopper-m.-e.|Walker2d-m.-e.|
> |-|-|-|-|
> |CQL|62.4|98.7|111.0|
> |CQL+BC|85.7|111.8|104.3|
> |CQL+BC(s.)|91.8|111.2|109.2|
> |CQL+BC(s.a.)|88.3|111.4|108.9|
> |OSR($\beta$=0)|92.3|111.8|110.1|
> |OSR|**94.7**|**114.3**|**113.1**|
>
> First, we run our OSR algorithm with the noisy level being zero and compare it with CQL - an OSR baseline without the inverse dynamic model（IDM). The results indicate that our OSR ($\beta$=0) (the 5th row) outperforms CQL (the 1st row) significantly on the 3 tasks. This demonstrates the significance of the inverse dynamics model (IDM) in helping the agent to navigate to more safe regions. Similar performance improvement can be observed if we replace the IDM model with the behavior policy (see the CQL + BC setting, the second row), but its effect is not so significant as our OSR method.
>
> Then we introduce the state noise into our base model OSR ($\beta$=0). The results show that this further improves the performance of the OSR by 2-3$\%$ (last row), and adding noise improves the performance of CQL + BC (the 3rd and the 4th row) as well except on the Hopper task, while the proposed OSR($\beta$>0) method consistently improves the performance over all the three environments.
>
> **Q2:... normalizing the states ... useful?...**
>
> **Response:** We added the suggested normalization step into our implementation while keeping a consistent $\beta$ value. The results shown below reveal that overall normalizing the states is beneficial to the performance and is useful in setting the value of $\beta$ across different state dimensions (the dimension of the tested environments is 17,11,17, respectively).
>
> | |Halfcheetah-m.-e.|Hopper-m.-e.|Walker2d-m.-e.|
> |-|-|-|-|
> |OSR|94.7|**114.3**|113.1|
> |OSR-norm|**95.8**|113.5|**113.8**|
>
> **Q3:... additional subsection ... on prior works....**
>
> **Response:** We will do that in the revised manuscript.  Our method can be used to improve the robustness of the agent against unfamiliar states, which is consistent with the goal of prior works in robust RL and robust offline RL, although in different manners.
>
> **Q4:... compare prior robust offline RL work....**
>
> **Response:** Per your suggestion, we have compared our method with RORL on the out-of-sample MuJoCo benchmark, and the results are given below; for comparison, we also give the results obtained without using the out-of-samples on these benchmarks. From the table, we observe that RORL generalizes well at out-of-sample states with slight noise, but its performance may suffer a lot under large perturbation. For example, while its normalized score on Halfcheetah-OOS-s. is 103.8, its performance drops significantly to 55.6 on the more challenging Halfcheetah-OOS-l. benchmark. On the contrary, our method performs much more stable across all the environments. In the attached PDF file, we give some visualization of the trajectories generated by both methods in some OOS benchmarks.
>
> || Halfcheetah-OOS-s.|Halfcheetah-OOS-m.|Halfcheetah-OOS-l.|Hopper-OOS-s.|Hopper-OOS-m.|Hopper-OOS-l.|Walker2d-OOD-s.|Walker2d-OOS-m.|Walker2d-OOS-l.|
> |-|-|-|-|-|-|-|-|-|-|
> |RORL-OOS|103.8|79.8|55.6|111.5|89.5|66.5|117.8|82.2|46.5|
> |RORL-w/o OOS|107.8|107.8|107.8|121.2|121.2|121.2|112.7|112.7|112.7|
> |OSR-OOS|94.1|92.7|91.7|113.3|113.2|110.1|111.4|109.2|106.1|
> |OSR-w/o OOS|94.7|94.7|94.7|114.3|114.3|114.3|113.1|113.1|113.1|
>
> **Q5:... comparisons with prior offline RL works that utilize a reverse model....**
>
> **Response:** Firstly, we would like to emphasize the conceptual difference between the inverse dynamics model (IDM) and the reverse dynamics model (RDM) --  IDM $I(a|s,s')$ behaves more like a policy which gives the action distribution while RDM  $R(s|s',a)$ is a generative model which predicts the reverse state distribution. Hence if the dimensions of state space is much higher than that of the action space, it would not be easy to train a RDM model or generate/image a new state, and vice versa for the IDM model. In addition, the RDM model performs a counterfactual query on the possible cause of action,  which can be problematic with a large action space - by contrast, the IDM model directly infers the most likely action that leads to a given consequence.
>
> Per your suggestion, we have compared our method with ROMI [1]  - an offline RL method based on RDM,  and the results are given below. This shows that although, on average, both methods perform comparably across the environments (average score: 68.9 (OSR) vs. 68.2 (ROMI)), our method performs much better in most 'medium' and 'medium-expert' benchmarks (e.g., Halfcheetah-m.-e., Hopper-m., Hopper-m.e., Walker2d-m.e.) than ROMI, indicating the advantage of using IDM if the underlying dataset covering a portion of high-value areas. However, our method may suffer from very noisy datasets (e.g., Hopper-r.), in which case RDM works better. This suggests that it could be beneficial to combine the advantages of both models for an even more robust offline RL.
>
> | |Halfcheetah-m.|Halfcheetah-m.-r.|Halfcheetah-m.-e.|Halfcheetah-r.|Hopper-m.|Hopper-m.-r.|Hopper-m.-e.|Hopper-r.|Walker2d-m.|Walker2d-m.-r.|Walker2d-m.-e.|Walker2d-r.|
> |-|-|-|-|-|-|-|-|-|-|-|-|-|
> |ROMI|**49.1**|**47.0**|86.8|24.5|72.3|**98.1**|111.4|**30.2**|84.3|**109.7**|109.7|7.5|
> |OSR(ours)|48.8|46.8|**94.7**|**35.2**|**83.1**|96.7|**113.1**|10.3|**85.7**|87.9|**114.3**|**13.5**|
>
> [1] Wang J, Li W, Jiang H, et al. Offline reinforcement learning with reverse model-based imagination. Advances in Neural Information Processing Systems, 2021, 34: 29420-29432.

---

> > ### Comment · Reviewer_cN6c · 2023-08-11
> > **Thank you for the response**
> >
> > Thank you for providing such a detailed response. I particularly appreciate the additional ablation/comparison conducted and the clarification regarding the reverse model used in prior work, as these aspects enhance the persuasiveness of the paper.
> >
> > Regarding Q4, I find the results to be intriguing. However, I am curious about the specific injected perturbation scales employed for RORL and OSR during training. It would be greatly appreciated if the authors could provide more details on these experiments.

---

> > > ### Author Response · Authors · 2023-08-11
> > > **Than you for the comment**
> > >
> > > Dear Reviewer cN6c,
> > >
> > > Thank you for the comment. We peformed the experiments in the response of **Q4** with the following hyperparameters of RORL in training, and other hyperparameters are same as those introduced in [1], as well.
> > >
> > > |  |Halfcheetah-medium-expert|Hopper-medium-expert|Wakler2d-medium-expert|
> > > |-|-|-|-|
> > > |Scalar $\epsilon_{OOD}$ of OOD Loss|0.00|0.01|0.01|
> > > |Scalar $\epsilon_{Q}$ of Q Smooth Loss|0.001|0.005|0.01|
> > > |Scalar $\epsilon_{P}$ of Policy Smooth Loss|0.001|0.005|0.01|
> > >
> > > While we trained the proposed OSR using the hyperparameters shown in the table below, and you can achieve the similar results in the response of **Q4** with the provided hyperparameters.
> > >
> > > |  |Halfcheetah-medium-expert|Hopper-medium-expert|Wakler2d-medium-expert|
> > > |-|-|-|-|
> > > |Scalar $\beta$ of noise injection| 1e-3|5e-3|2e-3|
> > > |Weight $\lambda$ of OSR term|0.5|0.5|0.5|
> > > |Weight $\alpha$ of CQL term|10.0|5.0|5.0|
> > >
> > > [1] Yang R, Bai C, Ma X, et al. Rorl: Robust offline reinforcement learning via conservative smoothing[J]. Advances in Neural Information Processing Systems, 2022, 35: 23851-23866.

---

> > > > ### Comment · Reviewer_cN6c · 2023-08-15
> > > > **About the noise scale**
> > > >
> > > > Thank you for the response. I notice that RORL employs larger noise scales in the adversarial experiments, as indicated in Table 5 of the RORL paper. Considering the sensitivity of OSR to the noise scale, as demonstrated in Appendix D.1, it would be more compelling to compare OSR-10 with RORL, both utilizing a larger noise scale. This comparison would help address the question of whether a larger noise scale can indeed improve the robustness to perturbations.

---

> > > > > ### Author Response · Authors · 2023-08-17
> > > > > **Adding experiments about the noise scale**
> > > > >
> > > > > Dear Reviewer cN6c,
> > > > >
> > > > > Thank you for the comment. Per your suggestion, we have trained our method OSR-10 based on the large perturbation scales in the following table (same as Table 5 in [1]), referred as OSR-10-large-noise (OSR-10-l.), on the three 'medium' MuJoCo datasets, while OSR-10 and RORL utilize the perturbation with smaller scales metioned before.
> > > > >
> > > > > ||Halfcheetah-medium|Hopper-medium|Walker2d-medium|
> > > > > |-|-|-|-|
> > > > > |Scalar $\epsilon_{OOD}$ of OOD Loss|0.00|0.02|0.03|
> > > > > |Scalar $\epsilon_{OSR}$ of OSR Loss|0.05|0.005|0.07|
> > > > >
> > > > > Then we run OSR-10-l. and RORL-l. (RORL trained on the perturbation scales in the Table 5 of [1] on 'medium' as well) in both the adversarial environments introduced in [1] and the OOS MuJoCo benchmarks proposed in our paper. The results of both methods in the adversarial environments are listed in the table below, where we set the scale of perturbation of the adversarial environments as 0.15, and we also provide the two methods' results in the clean environments for comparison (3rd and 6th rows).
> > > > >
> > > > > The results show that the peformance of RORL and OSR-10 improves by 13.1% and 12.1% respectively afer adding larger scale of perturbation in training. This indicates that increasing the difficulty of training data to some degree is indeed helpful in enhancing the generalization capbility of the learned agent, hence improving its robustness against adversarial attacks. It's our future work to further investigate the performance boundary of applying this trick.
> > > > >
> > > > > || Halfcheetah+random|Halfcheetah+action diff|Halfcheetah+min Q|Hopper+random|Hopper+action diff|Hopper+min Q|Walker2d+random|Walker2d+action diff|Walker2d+min Q|
> > > > > |-|-|-|-|-|-|-|-|-|-|
> > > > > |RORL-OOS|53.4|51.7|43.9|67.8|62.3|41.3|76.7|76.4|68.6|
> > > > > |RORL-l.-OOS|58.4|52.1|46.4|82.8|77.3|46.2|93.1|89.2|72.5|
> > > > > |RORL-l.-w/o OOS|61.2|61.2|61.2|102.8|102.8|102.8|97.4|97.4|97.4|
> > > > > |OSR-10-OOS|55.2|53.3|50.6|70.6|63.8|44.9|79.4|79.1|78.9|
> > > > > |OSR-10-l.-OOS|59.3|54.6|55.0|87.9|76.2|49.4|91.7|92.8|81.6|
> > > > > |OSR-10-l.-w/o OOS|63.3|63.3|63.3|103.5|103.5|103.5|100.7|100.7|100.7|
> > > > >
> > > > > Besides, the results of both methods in the OOS MuJoCo benchmarks are listed in the table below. From the results (1st vs. 2nd and 3rd vs. 4th), we observe the similar phenomenon as before, that is, larger scale of perturbation in training is benificial to the robustness, with 7.5% and 2.4% improvement for RORL and OSR-10 repsectively.
> > > > >
> > > > > || Halfcheetah-OOS-s.|Halfcheetah-OOS-m.|Halfcheetah-OOS-l.|Hopper-OOS-s.|Hopper-OOS-m.|Hopper-OOS-l.|Walker2d-OOD-s.|Walker2d-OOS-m.|Walker2d-OOS-l.|
> > > > > |-|-|-|-|-|-|-|-|-|-|
> > > > > |RORL-OOS|55.3|47.6|35.4|100.4|94.4|82.1|92.9|86.5|71.8|
> > > > > |RORL-l.-OOS|59.8|55.8|42.3|101.1|97.5|88.6|92.7|89.2|77.3|
> > > > > |OSR-10-OOS|59.4|56.5|50.8|100.8|98.3|94.7|92.4|90.3|88.6|
> > > > > |OSR-10-l.-OOS|61.3|59.7|53.3|101.3|99.2|97.6|92.6|91.3|90.7|
> > > > >
> > > > > [1] Yang R, Bai C, Ma X, et al. Rorl: Robust offline reinforcement learning via conservative smoothing[J]. Advances in Neural Information Processing Systems, 2022, 35: 23851-23866.

---

> > > > > > ### Comment · Reviewer_cN6c · 2023-08-18
> > > > > > **Thanks for your response and additional results**
> > > > > >
> > > > > > Thank you for your response! I sincerely appreciate the authors for conducting such comprehensive experiments within the limited rebuttal period. The new results are particularly significant as they demonstrate that (1) both OSR-10 and RORL can enhance their robustness by increasing the perturbation scale to a certain extent, and (2) OSR-10-l outperforms the previous approach RORL by a considerable margin in both observation perturbation benchmarks and OOS MuJoCo benchmarks.
> > > > > >
> > > > > > These findings effectively address my concerns, and I am now inclined to increase my score to 7. The authors are expected to include these new results during the rebuttal phase in the revision.

---

> > > > > > > ### Author Response · Authors · 2023-08-18
> > > > > > > **Thank you for raising the score**
> > > > > > >
> > > > > > > Thank you for raising the score. We appreciate your valuable feedback on improving our work. If you have any other questions, please post them and we are happy to continue our communication.

---

### Official Review · Reviewer_kCan · 2023-07-03

**Soundness:** 2 fair
**Presentation:** 3 good
**Contribution:** 2 fair
**Rating:** 5
**Confidence:** 4

**Summary:**

The paper proposes a solution to the problem of state distributional shift in offline RL - the agent takes unreliable actions in out-of-sample states during testing. The paper introduces the use of inverse dynamics models to guide the state recovery behavior of learned policy. Without constructing forward model, OSR aligns the learned policy’s transited state distribution at out-of-sample state with the offline dataset. Experimental results demonstrate the effectiveness of the proposed method.

**Strengths:**

I think the idea and effort to deal with state deviation in offline RL is meaningful and promising.

The paper is easy to read and understand.

The experimental results in the Out-of-sample MuJoCo setting are interesting and demonstrate the advantage of training state recovery policy.

**Weaknesses:**

It seems there is a non-negligible difference between the proposed method (theory) and the implementation. Eq. 7 and Eq.11 are not equivalent since the expectation w.r.t. s' is put outside the KL in Eq.11. One can find that they are not equal after simple calculation. I suppose this difference eliminates the need of predicting forward dynamics.

The paper lacks sufficient ablation study to support the effectiveness of the OSR/OSR-v term.

**Questions:**

How does OSR (not OSR-v) perform if the CQL term is removed? I think the OSR regularization term in Eq. 11 implies a kind of behavior cloning. Can this term alone suppress extrapolation error and overestimation?

In Fig. 6, it seems that the weight $\lambda$ for the OSR regularization has little effect on the performance?

**Limitations:**

Although the performance in Tab. 1 is good, the hyperparamters need to be tuned per dataset (Tab. 3,4).

---

> ### Author Rebuttal · Authors · 2023-08-08
>
> Thanks for your thoughtful comments. We provide clarification to your questions as below. We appreciate it if you have any further questions or comments.
>
> **W1:It seems there is a non-negligible difference between the proposed method (theory) and the implementation. Eq. 7 and Eq.11 are not equivalent since the expectation w.r.t. s' is put outside the KL in Eq.11. One can find that they are not equal after simple calculation. I suppose this difference eliminates the need of predicting forward dynamics.**
>
> **Response:** Thank you for the comment. We will make the derivation of Eq.11 from Eq.7 more clear in the revised manuscript. In particular,  a sample from the mixed dataset $\mathcal D_{tot}$ is denoted as $(\tilde{s}, a, r, s')$, where  $\tilde{s}$ can be either the original $s$ or a perturbed one $\hat{s}$ (according to Eq.5 ).
>
> To implement Eq.7, we remove the expectation w.r.t. s' inside the KL using Monte Calro approximation with the sample number N as 1, i.e.,
>
> $\min_\pi\mathbb E_{\tilde{s}\sim\mathcal D_{tot}}\Big[KL\bigg(\mathbb E_{s'\sim P(s'|s,\pi_\beta)}I^{\pi_\beta}
> (a|\tilde{s},s')\bigg\|\pi(a|\tilde{s}) \bigg) \Big|s, \pi_\beta\Big]$
>
> $\approx\min_\pi\mathbb E_{\tilde{s}\sim\mathcal D_{tot}}KL\Big(\frac{1}{N}\sum^N_iI^{\pi_\beta}(a|\tilde{s},s'_i)\Big\|\pi(a|\tilde{s})\Big)$
>
> $\approx\min_\pi\mathbb E_{\tilde{s}\sim\mathcal D_{tot}}KL\Big(I^{\pi_\beta}(a|\tilde{s},s')\Big\|\pi(a|\tilde{s})\Big)$
>
> Therefore, more strictly, Eq.11 should be written as:
>
> $L_{sr} = E_{\tilde{s}\sim\mathcal D_{tot}}KL\Big( I^{\pi_\beta}(a|\tilde{s},s')\Big\|\pi(a|\tilde{s})\Big)$
>
> where $\tilde{s}, s'$ are in the same couple sampled from the mixed dataset $\mathcal D_{tot}$. It should be emphasized that we do not calculate the expectation of $s'$ outside of KL.
>
> **Q1:How does OSR (not OSR-v) perform if the CQL term is removed? I think the OSR regularization term in Eq. 11 implies a kind of behavior cloning. Can this term alone suppress extrapolation error and overestimation?**
>
> **Response:** This is an interesting question, and per your suggestion, we have conducted a more comprehensive ablation study on three environments, shown in the Table below, where each row gives the normalized scores of a specific compared ablated method.
>
> | |Halfcheetah-m.-e.|Hopper-m.-e.|Walker2d-m.-e.|
> |-|-|-|-|
> |QL|9.8|0.3|0.2|
> |QL+BC|41.2|44.7|73.6|
> |QL+IDM|47.5|53.9|80.2|
> |CQL|62.4|98.7|111.0|
> |CQL+BC|85.7|111.8|104.3|
> |CQL+IDM(OSR)|**94.7**|**114.3**|**113.1**|
>
> Based on the above results, we have the following observations:
>
> 1) the IDM model is useful - we can see that the performance ranking is: OSR > CQL+BC > CQL , where BC denotes traditional behavior cloning (using behavior policy). The IDM can be thought of as a behavior cloning but taking the consequence of an action into consideration when cloning that action.
>
> 2) regulating the Q function search space is important for the generalization capability of an offline RL agent - note that if we replace CQL with standard QL in OSR, we have the following performance ranking: OSR>QL+IDM>QL+BC, which shows that the CQL-stlye value function learning is important both for IDM and BC. One possible reason for this is that the agent is trained under the actor-critic framework where both value function and policy play a role, and most importantly, in the setting of offline RL, conservative learning like CQL is critical for suppressing extrapolation error and overestimation, as pointed out by the reviewer.
>
> 3) IDM and CQL are complementary to each other - the IDM provides a way to guide the agent to navigate to safer regions, which allows the agent to learn more smart behavior when encountering unfamiliar states - in the sense that the desired behaviors of an agent should be rational (i.e., being less likely to be punished by the objective function of CQL ) not only under normal in-distribution states but under difficult unseen situations as well (this latter point is less studied in current literature) . The IDM can also be thought of as a mechanism to control the training procedure of CQL, such that it will behave less  'overly-conservatively' during learning, potentially improving its generalization capability.
>
> **Q2:In Fig. 6, it seems that the weight $\lambda$ for the OSR regularization has little effect on the performance?**
>
> **Response:** Thank you for the comment. The reason why this happens may be that the range of the $\lambda$s we choose is too small. To further evaluate how the hyperparameters $\lambda$ and $\beta$ affect the performance, we have attached more results of sensitive analysis based on the normalized score metrics as follows,
>
> Halfcheetah:
> |$\lambda$\\\\ $\beta$|1e-5|1e-4|1e-3|1e-2|1e-1|
> |-|-|-|-|-|-|
> |0(CQL)|62.4|62.4|62.4|62.4|62.4|
> |0.01|58.7|59.3|64.6|62.1|46.5|
> |0.1|54.2|76.4|83.4|63.7|50.3|
> |0.5|75.4|87.9|**94.6**|82.3|33.6|
> |1.0|73.2|89.2|92.4|46.7|34.4|
> |10.0|64.7|67.9|76.5|43.9|32.7|
>
> Hopper:
> |$\lambda$\\\\ $\beta$|1e-5|1e-4|1e-3|1e-2|1e-1|
> |-|-|-|-|-|-|
> |0(CQL)|111.0|111.0|111.0|111.0|111.0|
> |0.01|109.3|110.2|111.6|46.3|20.6|
> |0.1|111.4|112.1|111.3|29.1|18.7|
> |0.5|111.5|112.3|112.9|17.1|20.4|
> |1.0|112.1|111.7|**113.0**|17.4|14.5|
> |10.0|98.3|70.8|69.6|22.6|13.3|
>
> Walker2d:
> |$\lambda$\\\\ $\beta$|1e-5|1e-4|1e-3|1e-2|1e-1|
> |-|-|-|-|-|-|
> |0(CQL)|98.7|98.7|98.7|98.7|98.7|
> |0.01|101.1|102.5|104.6|94.1|33.3|
> |0.1|103.2|108.9|112.4|97.6|16.6|
> |0.5|109.4|112.6|**114.1**|83.3|16.8|
> |1.0|108.2|110.1|113.8|84.7|20.1|
> |10.0|101.7|100.8|89.1|72.8|17.1|
>
> From the results, we observe that we should be cautious in choosing a $\beta$ that is not very large; otherwise, it could lead to failure. We remark that it is better to choose the appropriate hyperparameters in the neighborhood of the bold data listed in the table above.

---

> > ### Comment · Reviewer_kCan · 2023-08-16
> > **About my first concern**
> >
> > Thank you very much for the detailed reply. About your response to my first concern:
> >
> > >"we remove the expectation w.r.t. s' inside the KL using Monte Calro approximation with the sample number N as 1."
> >
> > In my understanding, it is biased for the Monte Calro sampling to approximate the expectation within the non-linear function KL.
> >
> > Besides, to make the derivation more clear, we can show the dependence between $\tilde{s}$ and $s'$. We can add the original $s$ to the tuple $(\tilde{s},s')$ in the $D_{tot}$:  $(s,\tilde{s},s')$, where $s$ is the state before perturbation.
> >
> > In my understanding, the sample based minimization of Eq. 7 should be
> >
> > $\mathbb{E} _{(s,\tilde{s})\sim D _{tot}} KL \left( \mathbb{E} _{s' \sim D _{tot} (\cdot| s,\tilde{s})}I^{\pi _\beta}(a|\tilde{s},s') | \pi(a|\tilde{s})\right)$
> >
> > However, because the dataset stores coupled $(s,\tilde{s},s')$, the algorithm (Eq. 11) actually takes the following sample based optimization:
> >
> > $\mathbb{E} _{(s,\tilde{s},s') \sim D _{tot}} KL \left( I ^{\pi _\beta}(a | \tilde{s},s') | \pi(a|\tilde{s})  \right) = \mathbb{E} _{(s,\tilde{s})\sim D _{tot}} \mathbb{E} _{s' \sim D _{tot}(\cdot| s,\tilde{s})} KL \left(I ^{\pi _\beta}(a|\tilde{s},s') | \pi(a|\tilde{s})  \right)$
> >
> > The above two equations are not equivalent. More clarification would be appreciated. Thanks.

---

> > > ### Author Response · Authors · 2023-08-17
> > > **More clarification about your first concern**
> > >
> > > Thank you for your comment. First, per your suggestion, we would add the original $s$ to the tuple $(\tilde{s}, s')$ in the $D_{tot}$ for clearer derivation, as $(s, \tilde{s}, s')$. And we have also noticed your concern - it looks like that the proposed optimization objective (Eq.7) is not equivalent to its actual implementation (Eq.11), i.e.,
> > >
> > > $\mathbb E_{(s,\tilde{s})\sim D_{tot}} D_{KL}\bigg[\mathbb E_{s'\sim P(s'|s, \pi_\beta)} I^{\pi_\beta}(a|\tilde{s}, s')\bigg\|\pi(a|\tilde{s})\bigg] \neq\mathbb E_{(s,\tilde{s})\sim D_{tot}}\mathbb  E_{s'\sim P(s'|s, \pi_\beta)}D_{KL}\bigg[I^{\pi_\beta}(a|\tilde{s}, s')\bigg\|\pi(a|\tilde{s})\bigg]$, where $P(s'|s, \pi_\beta)$ is the transition distribution, denoted as $D_{tot}(s'|s, \tilde{s})$ as well for sampled version.
> > >
> > > However, we would like to clarify that these two optimization problems are actually equivalent, in the sense that they induce the same solution,
> > >
> > > $\arg\min\limits_\pi \mathbb E_{(s,\tilde{s})\sim D_{tot}} D_{KL}\bigg[\mathbb E_{s'\sim P(s'|s, \pi_\beta)} I^{\pi_\beta}(a|\tilde{s}, s')\bigg\|\pi(a|\tilde{s})\bigg] = \arg\min\limits_\pi \mathbb E_{(s,\tilde{s})\sim D_{tot}}\mathbb  E_{s'\sim P(s'|s, \pi_\beta)}D_{KL}\bigg[I^{\pi_\beta}(a|\tilde{s}, s')\bigg\|\pi(a|\tilde{s})\bigg]$ (1)
> > >
> > > In what below, we give the detailed derivation,
> > >
> > > $\arg\min\limits_\pi \mathbb E_{(s,\tilde{s})\sim D_{tot}} D_{KL}\bigg[\mathbb E_{s'\sim P(s'|s, \pi_\beta)} I^{\pi_\beta}(a|\tilde{s}, s')\bigg\|\pi(a|\tilde{s})\bigg]$ \ \ \ \ (2)
> > >
> > > $= \arg\min\limits_\pi \mathbb E_{(s,\tilde{s})\sim D_{tot}}\sum_a \mathbb  E_{s'\sim P(s'|s, \pi_\beta)}I^{\pi_\beta}(a|\tilde{s}, s')\log\frac{\mathbb E_{s''\sim P(s''|s, \pi_\beta)} I^{\pi_\beta}(a|\tilde{s}, s'')}{\pi(a|\tilde{s})}$  \ \ \ \ (3)
> > >
> > > $= \arg\min\limits_\pi \mathbb E_{(s,\tilde{s})\sim D_{tot}}\mathbb  E_{s'\sim P(s'|s, \pi_\beta)}\sum_a I^{\pi_\beta}(a|\tilde{s}, s')\log\frac{\mathbb E_{s''\sim P(s''|s, \pi_\beta)} I^{\pi_\beta}(a|\tilde{s}, s'')}{\pi(a|\tilde{s})}$  \ \ \ \ (4)
> > >
> > > $= \arg\min\limits_\pi \mathbb E_{(s,\tilde{s})\sim D_{tot}}\mathbb  E_{s'\sim P(s'|s, \pi_\beta)}\sum_a I^{\pi_\beta}(a|\tilde{s}, s')\log\mathbb E_{s''\sim P(s''|s, \pi_\beta)} I^{\pi_\beta}(a|\tilde{s}, s'') + \mathbb E_{(s,\tilde{s})\sim D_{tot}}\mathbb E_{s'\sim P(s'|s, \pi_\beta)}\sum_a I^{\pi_\beta}(a|\tilde{s}, s')\log\frac{1}{\pi(a|\tilde{s})}$  \ \ \ \ (5)
> > >
> > > Note the term $\mathbb E_{(s,\tilde{s})\sim D_{tot}}\mathbb  E_{s'\sim P(s'|s, \pi_\beta)}\sum_a I^{\pi_\beta}(a|\tilde{s}, s')\log\mathbb E_{s''\sim P(s''|s, \pi_\beta)} I^{\pi_\beta}(a|\tilde{s}, s'')$ in Eq.(5) is a constant w.r.t. $\pi$, hence we can remove it as follows,
> > >
> > > $(5)= \arg\min\limits_\pi \mathbb E_{(s,\tilde{s})\sim D_{tot}}\mathbb  E_{s'\sim P(s'|s, \pi_\beta)}\sum_a I^{\pi_\beta}(a|\tilde{s}, s')\log\frac{1}{\pi(a|\tilde{s})}$  \ \ \ \ (6)
> > >
> > > We remark that $\mathbb E_{(s,\tilde{s})\sim D_{tot}} \mathbb E_{s'\sim P(s'|s, \pi_\beta)}\sum_a I^{\pi_\beta}(a|\tilde{s}, s')\log I^{\pi_\beta}(a|\tilde{s}, s')$ is a constant w.r.t. $\pi$, so we can add it onto Eq.(6) as follows,
> > >
> > > $(6)= \arg\min\limits_\pi \mathbb E_{(s,\tilde{s})\sim D_{tot}}\mathbb  E_{s'\sim P(s'|s, \pi_\beta)}\sum_a I^{\pi_\beta}(a|\tilde{s}, s')\log\frac{1}{\pi(a|\tilde{s})} + \mathbb E_{(s,\tilde{s})\sim D_{tot}}\mathbb E_{s'\sim P(s'|s, \pi_\beta)}\sum_a I^{\pi_\beta}(a|\tilde{s}, s')\log I^{\pi_\beta}(a|\tilde{s}, s')$  \ \ \ \ (7)
> > >
> > > $= \arg\min\limits_\pi \mathbb E_{(s,\tilde{s})\sim D_{tot}}\mathbb  E_{s'\sim P(s'|s, \pi_\beta)}\sum_a I^{\pi_\beta}(a|\tilde{s}, s')\log\frac{I^{\pi_\beta}(a|\tilde{s}, s')}{\pi(a|\tilde{s})}$  \ \ \ \ (8)
> > >
> > > $= \arg\min\limits_\pi \mathbb E_{(s,\tilde{s})\sim D_{tot}}\mathbb  E_{s'\sim P(s'|s, \pi_\beta)}D_{KL}\bigg[I^{\pi_\beta}(a|\tilde{s}, s')\bigg\|\pi(a|\tilde{s})\bigg]$  \ \ \ \ (9)
> > >
> > > And then we can remove the expectation w.r.t. $s'$ in Eq.(9) with Monte Calro approximation with the sample number $N$ as 1,
> > >
> > > $\arg\min\limits_\pi \mathbb E_{(s,\tilde{s})\sim D_{tot}}\mathbb  E_{s'\sim P(s'|s, \pi_\beta)}D_{KL}\bigg[I^{\pi_\beta}(a|\tilde{s}, s')\bigg\|\pi(a|\tilde{s})\bigg]$
> > >
> > > $\approx \arg\min\limits_\pi \mathbb E_{\tilde{s}\sim D_{tot}}\frac{1}{N}\sum_{i=1}^ND_{KL}\bigg[I^{\pi_\beta}(a|\tilde{s}, s'_{i})\bigg\|\pi(a|\tilde{s})\bigg]$  \ \ \ \ (10)
> > >
> > > $\approx \arg\min\limits_\pi \mathbb E_{\tilde{s}\sim D_{tot}}D_{KL}\bigg[I^{\pi_\beta}(a|\tilde{s}, s')\bigg\|\pi(a|\tilde{s})\bigg]$  \ \ \ \ (11)
> > >
> > > where $(s, \tilde{s}, s')$ are in the same couple sampled from the mixed dataset $D_{tot}$. Here the Eq.(11) is the Eq.11 in our paper.

---

> > > ### Author Response · Authors · 2023-08-22
> > > **Dear Reviewer kCan**
> > >
> > > Dear Reviewer kCan,
> > >
> > > 	Thank you for your efforts in improving our work. Our newest official commnet raised has provided more clarification about your first concern with a detailed derivation. We would be happy to include the above discussions in our work and we would also be very grateful if you could respond to these points.

---

> > > > ### Comment · Reviewer_kCan · 2023-08-22
> > > >
> > > > Thank you for your clarification and sorry for the delay. It is strongly recommended to add these to the appendix as it is not obvious. Besides, Line 212 should be revised, since they induce the same solution but the equations themselves are not equivalent. My main concern is addressed, hence I have raised my score.

---

> > > > > ### Author Response · Authors · 2023-08-22
> > > > > **Thank you for raising the score.**
> > > > >
> > > > > Thank you for raising the score. We appreciate your valuable feedback on improving our work and the discussions during the rebuttal phase would be included in the revision.

---

### Official Review · Reviewer_tm1v · 2023-07-06

**Soundness:** 3 good
**Presentation:** 3 good
**Contribution:** 3 good
**Rating:** 5
**Confidence:** 3

**Summary:**

The paper addresses the issue of state distributional shift in offline reinforcement learning, where an agent tends to take unreliable actions when faced with unseen states during testing. The authors propose a solution to encourage the agent to follow the state recovery principle when making decisions. In addition to considering long-term return, the agent takes into account the immediate consequences of its current action, prioritizing actions that are capable of recovering the state distribution of the behavior policy. To achieve this, the authors train an inverse dynamics model, which is then used to guide the state recovery behavior of the new policy. The authors demonstrate the effectiveness and feasibility of their approach by achieving state-of-the-art performance on general offline RL benchmarks. Importantly, the proposed method aligns the transited state distribution of the new policy with the offline dataset at out-of-sample states without the need for explicit prediction, which is particularly challenging in complex and high-dimensional environments.

**Strengths:**

1. The paper addresses the state distributional shift problem, which has been relatively overlooked in prior research that primarily concentrates on mitigating out-of-distribution (OOD) actions during training.

2. The paper exhibits a high level of writing proficiency, logical organization, and reader-friendliness.

3. The utilization of an inverse dynamics model as a guide for state recovery represents a novel approach in the field.

4. The paper provides a rigorous theoretical analysis that demonstrates the effectiveness of the proposed algorithm.

**Weaknesses:**

A significant concern pertains to the construction of the noisy dataset. Two subproblems arise in this context. Firstly, the constructed noisy states may not adequately reflect the distribution of out-of-sample states encountered in the real world. For instance, when dealing with a robot control task, simply introducing Gaussian noise to the logged states might not align well with practical out-of-sample states due to various physical or environmental constraints on state transitions. Secondly, the constructed transition data, where the noisy state transitions to the next state, may not accurately represent the transition distribution in the real world. Consequently, this discrepancy between the two sources of data conflicts with the fundamental claim of the paper, as it becomes challenging to demonstrate that the inverse dynamics learned truly reflect the correct dynamics when the data sources are mismatched. More discussion or experimental results should be presented to address this concern.

**Questions:**

In light of the aforementioned weaknesses, I have a few questions that, if addressed by the authors, could potentially enhance the overall quality of the paper:

1. How can the issue of constructing a noisy dataset be mitigated to better reflect the distribution of out-of-sample states encountered in real-world scenarios, considering the presence of physical and environmental constraints on state transitions?
2. Is there a way to ensure that the constructed transition data accurately represents the true transition distribution in real-world settings, thus aligning with the core claim of the paper regarding the learned inverse dynamics?
3. Could the authors provide further evidence or explanations to alleviate concerns regarding the mismatch between the constructed noisy dataset and the actual out-of-sample states, thereby strengthening the validity of the proposed algorithm?


I believe that addressing these questions would significantly contribute to improving the paper and potentially enhance its evaluation score.

**Limitations:**

One notable limitation of this work is the potential mismatch between the constructed transition distribution and the true transition distribution encountered in real-world scenarios. This disparity may restrict the practical applicability of the proposed method.

---

> ### Author Rebuttal · Authors · 2023-08-08
>
> Thank you for your comment. We appreciate your questions and provide clarification below.
>
> **Q1: ... noisy dataset be mitigated to better reflect the distribution of out-of-sample states ...?**
>
> **Response:** Before answering this concern, please note that modeling OOD samples is not our ultimate goal, and the reasons are as follows: 1) encountering out-of-sample situations is almost inevitable for any agent due to the presence of approximal errors and other factors (e.g., finite sample size),  2) there exists experimental evidence showing that even when the OOD states are not well modeled (e.g., using our naive Gaussian noise injection), state of art performance can still be achieved if we can find a way to properly guide the agent to navigate to more safe regions (e.g., using our inverse model). Please see our response to Reviewer tZMd for more details.
>
> Due to the above reasons, the focus of our work is actually not to explore the real distribution of out-of-sample states, but on how to improve the robustness of the agent against unfamiliar states, and to encourage the agent to learn more smart behavior when encountering such situations.
>
> As noise injection is a well-known method to enhance the robustness of the learning algorithm, per your suggestion,  we have explored alternative way to better model the OOD states, which takes into account the physical and environmental constraints, as well as the currently learned policy and Q-values. Particularly, we employ a series of adversarial attacks, including random, action difference, and minimum Q attacks, to generate OOD samples, and use them as data augmentation. The detailed results are given in our response to **W1** of Reviewer tZMd.
>
> From the results, we can see that the utilization of a more intricate and meticulously designed noisy dataset is indeed useful in enhancing the performance of our method, probably due to the fact that such methods can improve the efficiency of OOD sampling and expand the coverage of noisy dataset.
>
> **Q2: ... ensure that the constructed transition data accurately represents the true transition ...**
>
> **Response:** Thank you for your comment. Instead of treating it as a dynamic model in representing the true transition distribution, in our work, we regard the inverse model (IDM) as an extended policy. It takes the current state (which may be unseen) $s$ and the one-step target state $s'$ as inputs, providing guidance to the agent on how to reach $s'$ from $s$ as effectively as possible. Although there is no strict theoretical guarantee that the IDM accurately predicts the true transition distribution at unseen states in real-world settings, our experimental results, shown in Fig. 3, 10, 11, and 12 of our paper, demonstrated that the IDM, acting as an extended policy, does aid the agent to behave better under the unseen states.
>
> To further validate the opinion above, we conduct a simple experiment using MuJoCo suites. State pairs $(s, s')$ are sampled from the offline dataset, and out-of-sample states $\hat{s}$ are generated via adversarial attacks [1]. By evaluating the average distance between the target state $s'$ and the actual state $\hat{s}'$ obtained by taking actions from the IDM's policy $IDM(a|\hat{s}, s')$, we assess the IDM's state recovery ability. A comparison with a CQL policy is also performed, yielding the following results:
>
> | |Halfcheetah|Hopper|Walker2d|
> |-|-|-|-|
> |Distance($s$,$\hat{s}$)|12.23|3.41|7.75|
> |Distance($s'$, $\hat{s'}_{IDM}$)|**11.79**|**2.80**|**5.49**|
> |Distance($s'$, $\hat{s'}_{CQL}$)|17.51|3.52|9.08|
>
> From these results we can see that compared to traditional policies like CQL, the action guidance provided by IDM effectively reduce the difference between $s$ and $s'$. It is important to note that while IDM does not guarantee accurate prediction of the true transition distribution at unseen states, the information it provides still proves valuable in mitigating such risks.
>
> **Q3: ... concerns regarding the mismatch between the constructed noisy dataset and the actual out-of-sample states ...**
>
> **Response:** To assess the generalizability of our method, we need to first construct a dataset with more realistic out-of-sample states to test the learnt model. For this we employ a modified generative adversarial network (GAN), which is optimized as follows,
>
> $\min\limits_G \max\limits_D [E_{s\sim P(s)}[\log D(s)]+ E_{P_{G}(s')}[\log (1-D(s'))]] + \alpha\cdot E_{P_{G(s)}}H [\pi_\beta(\cdot|s)]$
>
> where $G$ is generator, $D$ is the discriminator, $P(s)$ is the real-word state distribution (the dataset) and $P_G(s)$ is the state distribution generated via $G$. $H$ is the entropy function. In words, the above objective aims to generate real-world samples (i..e., consistent with the in-distribution data) that confuse the behavior policy $\pi_\beta$ most; hence its output could be considered as kind of realistic out-of-sample states.
>
> In evaluation, we initialize the MuJoCo environments with the generated out-of-sample states and assess the generalizability of the learnt agent on them. The results, based on the average of normalized scores and recovery rate, are as follows:
>
> | |Halfcheetah|Hopper|Walker2d|
> |-|-|-|-|
> |CQL|20.4/33.8%|36.5/39.1%|13.1/12.6%|
> |OSR|**40.1/69.9%**|**72.0/88.8%**|**43.3/32.4%**|
>
> The above results indicate that our OSR effectively guides the agent to recover from most real-world out-of-sample situations (nearly 70%) in the Halfcheetah and Hopper tasks, despite the mismatch between the constructed (Gaussian) noisy dataset and the actual (GAN-based) out-of-sample states. Although the Walker2D task seems challenging to it, our OSR method performs significantly better than CQL, where a risk-guiding mechanism like ours is lacking.
>
> To gain further insights, we provide visualizations of typical out-of-distribution (OOD) states in the three tasks, along with the corresponding trajectories of OSR. Please refer to the uploaded PDF for the visualizations.

---

> > ### Comment · Reviewer_tm1v · 2023-08-14
> >
> > Thanks. I have improved my score.

---

> > > ### Author Response · Authors · 2023-08-14
> > > **Thanks for your comment.**
> > >
> > > Thank you for your comment. If you have any other questions, please post them. We are happy to continue our communication.

---

### Official Review · Reviewer_tZMd · 2023-07-08

**Soundness:** 3 good
**Presentation:** 3 good
**Contribution:** 2 fair
**Rating:** 6
**Confidence:** 4

**Summary:**

The authors tackle the state distributional shift problem in offline reinforcement learning, by learning to *recover* to states that are close to the in-distribution region, where the proposed method is named Out-of-sample State Recovery (OSR). They augment the offline dataset by generating new samples with Gaussian noise injected to states for training. Also, they train an inverse dynamics model (IDM) to predict the *recovering* action given the current and next actions. Based on it, they learn a policy and encourage it to imitate the IDM to output *recovering* actions using their KL divergence term, or penalize Q-values for going out-of-sample. The authors present empirical results in MuJoCo and AntMaze environments.

**Strengths:**

- The training of IDM with perturbed samples and its use for encouraging the policy to recover to seen states is novel to some extent.
- The experimental evaluation and analyses of the proposed methods are done in multiple settings and perspectives, which backs up the proposed methods empirically.
- State distributional shift is one of the problems that need to be tackled in the field of offline reinforcement learning.

**Weaknesses:**

- The noise injection may not be enough in more complex environments with more state dimensions, as the state space would be too big to cover possible close OOD states with random sampling.
- On a related note, as the size of the original offline dataset increases, the needed amount of augmented data could become too large or otherwise the augmentation might get less effective.
- In terms of presentation, I think merging Fig.13 and Fig.14 would be more informative and make the comparison of the results with OSR and OSR-v easier.

**Questions:**

- Do you think the proposed IDM-based approach is supposed to work better than existing pessimism approaches with dataset augmentation (using noise injection) in general, especially when no specific modifications to cause out-of-sample situations are made for the environment? If so, I would like to hear the justification.

**Limitations:**

The authors fairly covered the limitations of this work on the assumptions for the theoretical derivations and the recoverability depending on the type of out-of-sample situations.

---

> ### Author Rebuttal · Authors · 2023-08-08
>
> Thanks for your thoughtful comments. We provide clarification to your questions as below. We appreciate it if you have any further questions or comments.
>
> **W1: The noise injection ... not be enough in more complex environments with more state dimensions...**
>
> **Response**: We agree that Gaussian noise injection would quickly become challenging in modeling OOD in high dimensional state space. In our opinion, an ideal OOD sample generator should be both efficient  or purposeful and relatively insensitive to the dimensions of the state space. One possible way to achieve this is through adversarial attacks [1] which purposefully searches those OOD samples in each $s$ centered $\epsilon$-neighborhood such that they either cause large policy perturbation or have lower Q value. The method is also less sensitive to the dimensions of state space due to the technique of deep generative networks. We construct a new dataset using this method and denote the experimental results on it as OSR-a, shown below,
>
> | |Halfcheetah-m.|Halfcheetah-m.-r.|Halfcheetah-m.-e.|Halfcheetah-r.|Halfcheetah-e.|Hopper-m.|Hopper-m.-r.|Hopper-m.-e.|Hopper-r.|Hopper-e.|Walker2d-m.|Walker2d-m.-r.|Walker2d-m.-e.|Walker2d-r.|Walker2d-e.|
> |-|-|-|-|-|-|-|-|-|-|-|-|-|-|-|-|
> |OSR-a|**52.3**|**51.2**|**101.2**|**35.2**|**103.1**|**85.2**|**97.4**|112.7|**10.9**|112.2|**87.2**|86.8|113.4|**15.1**|**111.2**|
> |OSR|48.8|46.8|94.7|**35.2**|97.7|83.1|96.7|**113.1**|10.3|**113.1**|85.7|**87.9**|**114.3**|13.5|110.3|
>
> From the results we observe that OSR-a outperforms OSR at most benchmarks. We remark that better-designed out-of-sample generating methods would further improve the performance of our method, probably because such methods take into account the physical and environmental constraints, resulting in better sampling efficiency.
>
> **W2: ... dataset increases, the needed amount of augmented ...**
>
> **Reponse:** Please note that modeling OOD samples based on the observed experience is not our ultimate goal. Actually, as mentioned in our response to Q1 of Reviewer cN6c, the performance of our method is largely due to the inverse model, although noise injection is shown to further improve the performance - for your convenience, we copy the results below, where our method of OSR($\beta=0$) is roughly equivalent to CQL + IDM without noise injection, while OSR is with noise injection. From these we can see that it is IDM, instead of noise injection, that contributes most to the performance of OSR, and data augmentation is more like icing on the cake for OSR (see 2nd and 3rd rows). Hence we conclude that even with no noise injection at all, our method would preserve most of its advantages in training a rational agent in the offline RL setting, provided that the dataset is large enough to train a good inverse dynamics model (IDM) .
>
> | |Halfcheetah-m.-e.|Hopper-m.-e.|Walker2d-m.-e.|
> |-|-|-|-|
> |CQL|62.4|98.7|111.0|
> |OSR($\beta$=0)|92.3|111.8|110.1|
> |OSR|**94.7**|**114.3**|**113.1**|
>
> **W3: ... merging Fig.13 and Fig.14 ...**
>
> **Response:** Thanks for your suggestion and we will do that in the revised manuscript.
>
> **Q1: .... OSR ... better than existing pessimism approaches with dataset augmentation (using noise injection) .... justification.**
>
> **Response:**  Our answer is yes, and our justification is as follows,
>
> Although data augmentation is useful when the training data are not so representative, encountering out-of-sample situations is almost inevitable for any agent due to the presence of approximal errors and other factors.  Such unfortunate situations aggravate when the new policy makes unreliable decisions at unseen states, causing the agent to deviate from the offline dataset, and existing data augmentation-based approaches, e.g., RORL [1] and ROMI [2], seldom consider the problem of how to recover from such scenarios.
>
> To give some experimental evidence on this, we compare two popular data augmentation approaches, i.e., ROMI and RORL,  with our method on the standard MuJoCo benchmark in the table below, where OSR-10 is OSR implemented with 10 Q networks,  the same as RORL.
>
> | |Halfcheetah-m.|Halfcheetah-m.-r.|Halfcheetah-m.-e.|Halfcheetah-r.|Hopper-m.|Hopper-m.-r.|Hopper-m.-e.|Hopper-r.|Walker2d-m.|Walker2d-m.-r.|Walker2d-m.-e.|Walker2d-r.|
> |-|-|-|-|-|-|-|-|-|-|-|-|-|
> |ROMI|49.1|47.0|86.8|24.5|72.3|98.1|111.4|30.2|84.3|**109.7**|109.7|7.5|
> |RORL|66.8|61.9|107.8|**28.5**|104.8|102.8|112.7|**31.4**|**102.4**|90.4|121.2|21.4|
> |OSR(ours)|48.8|46.8|94.7|35.2|83.1|96.7|113.1|10.3|85.7|87.9|114.3|13.5|
> |OSR-10(ours)|**67.1**|**64.7**|**108.7**|28.1|**105.5**|**103.1**|**113.2**|30.2|102.0|93.8|**123.4**|**23.1**|
>
> These results show that although only a very naive noise injection mechanism is used for data augmentation in our OSR, its performs comparable or better than both ROMI and RORL in which more complex data augmentation techniques are involved. Besides, we also compare our method with RORL in the Out-of-sample MuJoCo benchmarks, and the results could be found in our response to the reviewer cN6c, showing that our IDM-based offline RL method OSR works much more stable than the state of the art data augmentation-based method (RORL) on the challenging OOD benchmarks.
>
> Furthermore, in the attached PDF file, we illustrate visually how our method enables the agent to recover from many out-of-sample situations on MuJoCo benchmarks.
>
> [1] Yang R, Bai C, Ma X, et al. Rorl: Robust offline reinforcement learning via conservative smoothing[J]. Advances in Neural Information Processing Systems, 2022, 35: 23851-23866.
>
> [2] Wang J , Li W , Jiang H ,et al.Offline Reinforcement Learning with Reverse Model-based Imagination[J].arXiv e-prints, 2021.DOI:10.48550/arXiv.2110.00188.

---

### Author Rebuttal · Authors · 2023-08-09

Thank you for all reviewers' thoughtful and constructive comments on our works discussing a significant but overlooked issue, state distribution shifting,  on offline reinforcement learning.

In summary, our response includes the following aspects:

1.**[Efficiency of noise injection and OOD sampling]** Providing additional experimental results with the adversarial methods onto our implementation, referred as OSR-a, and additiional ablation study including the removal of noise ($\beta=0$) and so on. The detailed results and analysis could be found in the response to Reviewer tZMd's **W1** and **W2** and Reviewer tm1v's **Q1**.

2.**[Validity of the use of IDM]** Providing additional experimental results to explore the behavior of the inverse dynamics model (IDM) we use and its guided policy, enhancing the validity of our work. The detailed results and analysis could be found in the response to Reviewer tm1v's **Q2, Q3** and the visualization in the attached PDF file.

3.**[Additional ablation study]** Providing a more comprehensive ablation study (sensitive analysis), clarifying the role of each component in our work. This could be found in the response to Reviewer kCan's **Q1,Q2** and Reviewer cN6c's **Q1**.

4.**[Additional comparison study]** Providing a further comparison study with related works, including ROMI, a representative offline RL algorithm with the reversed model, and RORL, a SOTA robust offline RL algorithm. The detailed results and analysis could be found in the response to Reviewer tZMd's **Q1**, Reviewer cN6c's **Q5**.

5.**[Performing a prior robust offline RL algormthm in OOS MuJoCo benchmarks]** Providing experimental results of RORL, a SOTA robust offline RL algorithm, in the proposed OOS MuJoCo benchmarks. The detailed results and analysis could be found in the response to Reviewer cN6c's **Q4** and the visualization in the attached PDF file.

6.**[Gap between theory and implementation]** Clarifying this concern in the response to Reviewer kCan's **W1**.

7.**[State normalization]** Providing experimental results of our implementation with the state normalization trick. The detailed results and analysis could be found in the response to Reviewer cN6c's **Q2**.

8.**[Format suggestions]** Including Reviewer tZMd's **W3** and Reviewer cN6c's **Q3**. We will do that in the revised manuscript.

We hope our response could address the reviewers' concerns. If you have any further questions, please post them. We are pleased to have further discussions.

---

### Decision · Program_Chairs · 2023-09-21

**Decision:**

Accept (poster)

**Comment:**

This paper addresses the distributional shift problem in offline RL.  The method is based on promoting the pursuit of state recovery principle when selecting actions, namely biasing towards actions that can recover the state distribution of the behavior policy.  Theoretical results are shown to well align the transited state distribution with the behavior dataset when a state is out of sample.  Experiments on MuJoCo and AntMaze demonstrate the superior performance of the proposed method.

This paper is well written, proposing a solid solution to an important problem.  All reviewers, and myself, agree that it is a good addition to the conference. We hope that the reviews are helpful for improving the paper.